**Investigation**

# A model of Hill-Robertson interference caused by purifying selection in a nonrecombining genome

Hannes Becher (iD) ,[1,*] Brian Charlesworth (iD) [2]

[1]Royal (Dick) School of Veterinary Science, The Roslin Institute, The University of Edinburgh, Midlothian EH25 9RG, UK
[2]School of Biological Sciences, Institute of Ecology and Evolution, The University of Edinburgh, Edinburgh EH9 3FL, UK

*Corresponding author: The Roslin Institute, Royal (Dick) School of Veterinary Science, The University of Edinburgh, Midlothian EH25 9RG, UK. Email: H.Becher@ed.ac.uk

A new approach to modeling the effects of Hill-Robertson interference on levels of adaptation and patterns of variability in a nonrecombining genome or genomic region is described. The model assumes a set of _L_ diallelic sites subject to reversible mutations between beneficial and deleterious alleles, with the same selection coefficient at each site. The assumption of reversibility allows the system to reach a stable statistical equilibrium with respect to the frequencies of deleterious mutations, in contrast to many previous models that assume irreversible mutations to deleterious alleles. The model is therefore appropriate for understanding the long-term properties of nonrecombining genomes such as Y chromosomes, and is applicable to haploid genomes or to diploid genomes when there is intermediate dominance with respect to the effects of mutations on fitness. Approximations are derived for the equilibrium frequencies of deleterious mutations, the effective population size that controls the fixation probabilities of mutations at sites under selection, the nucleotide site diversity at neutral sites located within the nonrecombining region, and the site frequency spectrum for segregating neutral variants. The approximations take into account the effects of linkage disequilibrium on the genetic variance at sites under selection. Comparisons with published and new computer simulation results show that the approximations are sufficiently accurate to be useful, and can provide insights into a wider range of parameter sets than is accessible by simulation. The relevance of the findings to data on nonrecombining genome regions is discussed.

Keywords: adaptation; genetic diversity; Hill-Robertson interference; linkage disequilibrium; purifying selection; site frequency spectrum

## Introduction

Population genomics data show that genomic regions with little or no genetic recombination usually show reduced levels of genetic diversity, as well as signs of reduced levels of adaptation such as reduced frequencies of optimal codons, reduced rates of fixation of selectively favorable nonsynonymous mutations, and increased frequencies of segregating nonsynonymous mutations (reviewed by Charlesworth and Campos 2014; Charlesworth and Jensen 2021). These patterns are consistent with the hypothesis that selection within genomes with low frequencies of recombination interferes with evolution at closely linked sites, a process commonly referred to as Hill-Robertson interference (HRI) (Hill and Robertson 1966; Felsenstein 1974). Several types of HRI have been modeled analytically, including the effects of the spread of beneficial mutations (selective sweeps) on closely linked selected sites (e.g. Kim 2004; Weissman and Barton 2012), and the elimination of strongly selected deleterious mutations (Charlesworth et al. 1993; Charlesworth 1994; Peck 1994; Johnson and Barton 2002). The study of other types of HRI, especially the case of multiple linked sites subject to weak purifying selection, when both selection and genetic drift play important roles, has relied heavily on computer simulations (Li 1987; McVean and Charlesworth 2000; Tachida 2000; Comeron and Kreitman 2002; Kaiser and Charlesworth 2009; Charlesworth et al. 2010; Seger et al. 2010; Hough et al. 2017; Devi et al. 2023; Daigle and Johri 2025). Given

that genomic regions with little or no recombination, and the whole genomes of asexual and highly inbreeding species, are likely to include many functionally significant nucleotide sites (both coding and noncoding) that are all closely linked to each other, HRI involving purifying selection seems likely to have major effects on variability and adaptation. Analytical results should help to provide a better understanding of these effects.

A recent paper has suggested a new approach to the problem of modeling HRI due to purifying selection in large nonrecombining genomic regions, providing formulae that seem to match the results of simulations over much of parameter space (Devi et al. 2023). In contrast to several previous analytical approaches to this problem (e.g. Santiago and Caballero 1998; Good et al. 2014; Santiago and Caballero 2016; Cvijovic et al. 2018; Melissa et al. 2022; Buffalo and Kern 2024; Strütt et al. 2025), reversible mutations were allowed between selectively favorable and deleterious mutations, permitting the establishment of a statistical equilibrium between mutation, selection, and drift. This is important, because the features of long-established low or zero recombination rate genomic regions should reflect their equilibrium properties, especially with respect to the effects of single base substitutions (Charlesworth et al. 2010). The formulae of Devi et al. (2023) were, however, derived by a heuristic argument based on clonal interference between beneficial mutations. Buffalo and Kern (2024) have developed an alternative approach to modeling the

effects of HRI on diversity at linked neutral sites, using the theoretical machinery developed by Santiago and Caballero (1998, 2016). However, their approach assumes unidirectional mutation from wild-type to deleterious alleles.

This paper re-examines the effects of HRI on multiple sites subject to purifying selection in the absence of recombination, by modifying the approach of Santiago and Caballero (1998, 2016) to allow for reversible mutations at selected sites, so that the system can reach a true stationary distribution of allele frequencies, allowing predictions of the mean frequencies of favored alleles at sites under selection. This provides a widely used measure of the overall level of adaptation (Li 1987; McVean and Charlesworth 2000; Comeron and Kreitman 2002; Charlesworth et al. 2010; Devi et al. 2023), which is described below.

In addition to developing an approximate formula for the level of adaptation at the sites under selection, we examine the properties of the gene tree for a block of linked loci under selection, allowing the pairwise diversity and site frequency spectrum (SFS) for neutral sites located among the selected sites to be calculated. We also show how the effect of HRI on the genetic variance at sites subject to selection, caused by the generation of linkage disequilibrium (LD), can be taken into account, to a good level of approximation.

For simplicity, a haploid genome is modeled; the results also apply to diploids, with minor changes in parameters, if the effects of mutations on fitness show intermediate dominance. Complications are, however, likely to arise in diploid populations where deleterious mutations are mostly partially recessive (Crow 1993), due to the possible effects of associative overdominance (Zhao and Charlesworth 2016; Becher et al. 2020; Gilbert et al. 2020; Charlesworth B 2022; Abu-Awad and Waller 2023), so that caution is needed in extending the conclusions to diploids. They should nevertheless be applicable to evolving Y or W chromosomes, which (subject to some caveats: Engleständter 2008) behave effectively as haploid for many purposes (Charlesworth and Charlesworth 2000; Olito et al. 2004), as well as U and V chromosomes in dioecious species with a haploid sexual stage in their lifecycle (Bull 1978; Charlesworth D 2022), some types of supernumerary chromosome (e.g. Torgasheva et al. 2019), clonally reproducing organisms (Price and Arkin 2015; Bast et al. 2018), and highly homozygous self-fertilizing populations, where recombination is effectively nearly absent (Charlesworth and Wright 2001; Burgarella et al. 2024; Daigle and Johri 2025).

## Methods
### The basic population genetics model

The basic model is identical to that used in theoretical studies of selection on codon usage, which assume a large number of nucleotide sites, each with 2 alleles ($A_1$ and $A_2$), with $A_1$ denoting the selectively favorable allele at a site and $A_2$ its deleterious alternative (Li 1987; Bulmer 1991; McVean and Charlesworth 1999; Tachida 2000). Carriers of $A_2$ have a fitness 1—$s$ relative to a fitness of 1 for $A_1$ and fitnesses are multiplicative across sites. The mutation rates for $A_1$ to $A_2$ and $A_2$ to $A_1$ are $u = \kappa v$ and $v$, respectively, where $\kappa$ is the mutational bias parameter. It is usually assumed that $\kappa > 1$, since preferred codons often end with GC (Hershberg and Petrov 2008; Sharp et al. 2010) and there is a nearly universal bias in favor of GC to AT base substitutions (Fu et al. 2011; Graur 2016, Chap.10). This model applies to any situation where selectively favored and deleterious alleles can be identified at individual nucleotide sites. Most analytical treatments assume complete

evolutionary independence among sites and use a single-locus population genetics model (Li 1987; Bulmer 1991; McVean and Charlesworth 1999).

The assumption of independence becomes increasingly implausible as linkage between sites becomes tighter (Li 1987; McVean and Charlesworth 2000; Comeron and Kreitman 2002). Here, we assume a genomic region of $L$ completely linked sites, with a total mutation rate to deleterious alleles of $U = Lu$. Assuming a Wright-Fisher population, in the absence of HRI the effects of genetic drift at each site are determined by a variance effective population size of $N_0$ (Crow and Denniston 1988). With allele frequencies $x$ and $1-x$ at a site, the new variance in allele frequency generated by random sampling from one generation to the next is $x(1-x)/N_0$. HRI can be viewed as reducing the effective population size ($N_e$) below this value (Hill and Robertson 1996; Santiago and Caballero 1998, 2016). While use of a modified effective population size does not capture all aspects of HRI, such as its effects on allele frequency spectra (Charlesworth et al. 1995; Santiago and Caballero 1998; O'Fallon, et al. 2010; Nicolaisen and Desai 2013; Good et al. 2014; Cvijovic et al. 2018), it is useful for understanding its effects on the fixation probabilities of new mutations and on pairwise diversity measures.

It may seem unnecessary to model reversible mutations when population genomic studies of normally recombining regions of genomes in organisms such as humans and *Drosophila* indicate that a majority of deleterious mutations are so strongly selected that they are kept at such low frequencies that reversion to wild-type can largely be ignored (e.g. Eyre-Walker and Keightley 2009; Kim et al. 2017; Johri et al. 2020). However, as shown by the results described below, the HRI effects of large numbers of completely linked sites can be so large that deleterious alleles reach intermediate frequencies or fixation, so that reversals cannot be ignored, as has also been found in previous studies (e.g. Charlesworth et al 2010; Devi et al. 2023).

Following McVean and Charlesworth (2000), Comeron and Kreitman (2002), and Devi et al. (2023), the effect of HRI on the level of adaptation can be quantified by the mean frequency of preferred alleles ($\bar{p}$) across the genomic region in question. At statistical equilibrium under drift, mutation and selection, $\bar{p}$ depends on the scaled selection coefficient $\gamma = 2N_e s$ and the mutational bias $\kappa$, and is determined to a good approximation by the Li-Bulmer equation (Li 1987; Bulmer 1991; McVean and Charlesworth 1999):

$$\bar{p} \approx \frac{1}{[1 + \kappa \exp(-\gamma)]} \tag{1a}$$

Strictly speaking, $\bar{p}$ is the proportion of sites that are fixed for the favorable allele $A_1$ among all fixed sites, and Equation (1a) is obtained by equating the rate of substitution of $A_2$ mutations arising at sites fixed for $A_1$ to the rate of substitution of $A_1$ variants arising at sites fixed for $A_2$ (Bulmer 1991), using the standard diffusion equation formula for the probability of fixation of a mutation in a finite population (Kimura 1964). However, this formula also accurately predicts the proportion of sites that are $A_1$ in state in a random sequence sampled from the population, provided that the proportion of sites that are segregating as opposed to being fixed is sufficiently small (McVean and Charlesworth 1999), so that the assumptions of the infinite sites model (Kimura 1971) are met.

Given an observed value of $\bar{p}$, $\gamma$ can be estimated as:

$$\hat{\gamma} = \ln\left(\frac{\kappa \bar{p}}{1 - \bar{p}}\right) \tag{1b}$$

This relation is often used to estimate the scaled strength of selection on codon usage from genomic data on the frequencies of preferred synonymous codons (e.g. Sharp et al. 2005, 2010).

If the value of the scaled selection coefficient in the absence of HRI, $\gamma_0 = 2N_0 s$, has been specified, the ratio $B = N_e/N_0$ when HRI operates can be equated to $\hat{\gamma}/\gamma_0$ (McVean and Charlesworth 2000). Devi et al. (2023) used the symbol $\phi$ instead of the more commonly employed $B$ for this quantity, and they termed the corresponding value of $N_e$ the "fixation effective population size," because Equation (1a) is derived in terms of fixation probabilities, as described above. The usefulness of Equation (1b) for estimating $\gamma$ breaks down when $\bar{p}$ is close to 1, so that $-\ln(1 - \bar{p})$ approaches infinity; this situation corresponds to $B\gamma_0 >> 1$, so that allele frequencies are close to deterministic mutation-selection balance at all sites. It is then impossible to obtain meaningful estimates of $B$ from $\bar{p}$ estimates. This is not a serious limitation in practice, since the interest in studying HRI comes from its effect in reducing $N_e$ below $N_0$.

## The effect of HRI on the level of adaptation
### The basic model
To use this modeling approach, we must be able to predict the fixation $N_e$. Santiago and Caballero (1998, 2016) showed that the effects of HRI under the situation considered here depend on the additive genetic variance in fitness ($V_a$). With haploidy, $V_a$ is equal to the total genetic variance in the absence of epistasis, as is assumed here, but the term additive genetic variance will be retained in order to make comparisons with diploid models. It is useful to start with the equation for the additive variance at a single selected site subject to reversible mutation and genetic drift, assuming that it can be described by the equation for free recombination, but, importantly, with a value of $N_e$ reflecting the effects of HRI. This differs from the approach of Santiago and Caballero (2016), Devi et al. (2023), and Buffalo and Kern (2024).

Using Equation (15) of McVean and Charlesworth (1999), modified for a haploid population, the expected nucleotide site diversity at such a site, under the infinite sites assumption that only a single mutation segregates at each variable site (Kimura 1971), is given by:

$$\pi_{sel} = \frac{2u}{s} \frac{[1 - \exp(-\gamma)]}{[1 + \kappa \exp(-\gamma)]} \tag{2a}$$

The corresponding variance in fitness at a site is equal to $\frac{1}{2}\pi_{sel}s^2$; the expected genic variance obtained by summing additive variances over all $L$ sites, giving a total mutation rate $U = Lu$, is thus equal to:

$$V_g = \frac{Us[1 - \exp(-\gamma)]}{[1 + \kappa \exp(-\gamma)]} \tag{2b}$$

It is important to note that Equation (2a) was derived by considering the expected contributions to diversity contributed by mutations traveling to fixation or loss, either $A_1$ to $A_2$ mutations or vice-versa, weighting their contributions by their respective rates of input into the population and probabilities of loss or fixation, respectively, as well as assuming that the majority of sites present at any one time are fixed for either $A_1$ or $A_2$ (McVean and Charlesworth 1999). The results derived below tacitly assume that the state of the population at any one time can be described by this expectation, which must of course be violated in practice, due to the highly stochastic nature of the processes involved.

They are likely, therefore, to be only approximate at best, and it is important to check them against simulations.

### The effect of LD on the additive genetic variance
In the following derivations, we assume that (1) the $N_e$ that appears implicitly in these equations can be used to predict $\bar{p}$ from Equation (1a) and (2) the contributions of LD to $V_a$ can be ignored, so that $V_a = V_g$. However, for a haploid genome $V_a = V_g + 2\sum_{i,j} C_{ij}$, where $C_{ij}$ is the covariance between sites $i$ and $j$ resulting from LD, and the sum is taken over all pairs of sites (Bulmer 1980, p. 158). Section 1 of the Appendix presents a heuristic argument, which shows that LD increases the diversity at selected sites, and hence $V_g$, by approximately the same amount that it contributes to a reduction in $V_a$. It follows that the relations derived below will cause the value of $\pi_{sel}$ to be underestimated, whereas the effect of HRI on $\bar{p}$ should be approximately correct. The problem of correcting for the effect of LD on the net additive genetic variance does not seem to have been considered previously in this context.

Finally, the multilocus simulations used by various investigators to evaluate the effects of HRI, including those used here, generally assume multiplicative fitness interactions among different sites. This raises the question of how to extrapolate to the whole genome from the single site equations described above. To do this, we note that the quantity used in the predictions of HRI effects is the additive genetic variance relative to the squared population mean fitness, $\bar{w}^2$ (e.g. Santiago and Caballero 2016). With multiplicative fitnesses and small effects of each site on fitness, we can use the well-known approximation for the total variance in fitness:

$$\frac{V(w)}{\bar{w}^2} \approx V(\ln w) \tag{3a}$$

With multiplicative fitnesses, the fitness of an individual is the exponential of $\ln(w)$, where $w$ is the sum of $1-s$ over each site whose state differs from wild-type. If each site has only a small effect, so that $\ln(w) \approx -s$, we then have

$$\frac{V(w)}{\bar{w}^2} \approx V_a(w) \tag{3b}$$

where $V_a(w)$ is the variance in fitness on the additive scale, obtained taking the variance of the sum of the effects of each site. This is the quantity that will be used here. As shown below, predictions based on this expression and the other approximations match the simulation results quite well.

### Calculating the effects of HRI
Using Equation (3) of Santiago and Caballero (2016) for the case of no recombination, the fixation $N_e$ is given implicitly by:

$$-\ln\left(\frac{N_e}{N_0}\right) = -\ln(B) = \frac{V_a^3}{V_m^2} \tag{4}$$

where $V_m$ is the increase in fitness variance per generation due to new mutations and $V_a \approx V_g$ is given by Equation (2b). In addition:

$$V_m = Us^2 \tag{5}$$

With $\gamma >> 1$ and no HRI between selected sites, the selected sites will equilibrate at deterministic mutation-selection balance,

such that the frequency of deleterious mutations at each site is $q^* = u/s << 1$ and $V_a \approx Us$, provided that $u << s$, as is usually the case. This case corresponds to the limiting value of Equation (2b) for large $\gamma$. Equations (4) and (5) then imply that:

$$B \approx \exp\left(-\frac{U}{s}\right) \qquad (6)$$

As expected, this is the classical background selection (BGS) formula for the reduction in $N_e$ caused by deleterious mutations in the absence of recombination (Charlesworth *et al.* 1993); it fails when HRI causes the frequencies of deleterious mutations to depart from their mutation-selection equilibrium values (Charlesworth *et al.* 1993; Santiago and Caballero 2016).

The challenge is thus to find an expression for $B$ when $\gamma$ is sufficiently small that HRI and genetic drift cause substantial departures from mutation-selection balance. This can be done by assuming that the $N_e$ involved in $\gamma = 2B\gamma_0$ in Equations (2) is determined by Equations (4) and (5), yielding the following expression:

$$-\ln(B) = \frac{U[1 - \exp(-\gamma)]^3}{s[1 + \kappa \exp(-\gamma)]^3} \qquad (7)$$

In general, this equation can only be solved for $B$ by numerical iteration. However, as well as the solution for the case of large $\gamma$ described above, a solution for small $\gamma$ and $B$ close to 1 can be found by assuming that $\ln(B) \approx -\delta$, where $\delta$ is sufficiently small that second-order terms in $\delta$ can be neglected. Writing $\gamma_0 = 2N_0s$ and $\alpha_0 = 2N_0U$ for the selection and mutation parameters scaled by twice the effective population size in the absence of HRI, the following approximation, which is valid only for small $\delta$, is then obtained:

$$\delta \approx \frac{X(1 + \kappa\gamma_0)}{(1 + 3X)} \qquad (8a)$$

where

$$X = \frac{\alpha_0^2\gamma_0^2}{(1 + \kappa)^3} \qquad (8b)$$

Given a value of $B$ from Equations (6)-(8), $\bar{p}$ can be obtained from Equation (1) and compared with the results of computer simulations of nonrecombining populations, providing a test of the validity of this approach. Devi *et al.* (2023) described the results of simulations of haploid genomes across a wide range of parameters, which yielded numerous estimates of $\bar{p}$. They also derived an approximate formula for $B$ that is valid for $N_es << 1$, large $L$ and $u/s << 1$. In the notation used here, it is equivalent to:

$$B \approx \left\{\frac{16(1 + \kappa)B_{max}}{\gamma_0\alpha_0}\right\}^{0.333} \qquad (9a)$$

where

$$B_{max} = \frac{1}{\left[1 + \dfrac{\gamma_0}{\ln\left(1 + \dfrac{\kappa s}{u}\right)}\right]} \qquad (9b)$$

Devi *et al.* (2023) stated that $B_{max}$ is the upper limit to $B$ when all sites are at mutation-selection balance (it is set equal to 1 if the numerator is < 1). As noted above, this is not biologically meaningful, since in this case the allele frequencies depend only on $u$ and $s$, so that $B_{max}$ has only a notional meaning, but is still useful for approximating the effects of HRI.

## Levels of neutral diversity under HRI

It is also interesting to determine the effect of purifying selection on variability at neutral sites embedded in a block of non-recombining sites, and this has been the subject of many previous theoretical studies (e.g. Charlesworth *et al.* 1993, 1995; Santiago and Caballero 1998, 2016; Comeron and Kreitman 2002; Gordo *et al.* 2002; Kaiser and Charlesworth 2009; O'Fallon *et al.* 2010; Seger *et al.* 2010; Zeng and Charlesworth 2011; Nicolaisen and Desai 2013; Good *et al.* 2014; Cvijovic *et al.* 2018; Melissa *et al.* 2022; Devi *et al.* 2023). Previous studies have shown that, when there are departures from mutation-selection balance, the value of $B$ for measuring the ratio of neutral diversity to its value in the absence of selection, denoted here by $B'$, tends to be larger than the $B$ determining the fixation probabilities of new mutations (Comeron and Kreitman 2002; Santiago and Caballero 2016; Devi *et al.* 2023). The corresponding value of $N_e$ for a haploid population is equivalent to the mean time to coalescence of a pair of alleles at a neutral site, so this quantity can usefully be referred to as the "coalescence effective population size" (Devi *et al.* 2023), denoted here by $N_e'$; Santiago and Caballero (2016) called it the "heterozygosity effective size."

$B'$ can be found as follows, using the equation at the top of p. 1270 of Santiago and Caballero (2016), which gives the rate of coalescence at generation $\tau$ back from the current generation as the reciprocal of an effective population size $N_e'(\tau)$, which is defined by:

$$\frac{1}{N_e'(\tau)} = \left(\frac{1}{N_0}\right)\exp\left(V_aQ_\tau^2\right) \qquad (10a)$$

where:

$$\begin{aligned}Q_\tau &= \sum_{x=0}^{\tau}\left(1 - \frac{V_m}{V_a}\right)^x = \frac{V_a}{V_m}\left[1 - \left(1 - \frac{V_m}{V_a}\right)^{\tau+1}\right] \\ &\approx \frac{V_a}{V_m}\left\{1 - \exp\left[-\frac{V_m}{V_a}(\tau + 1)\right]\right\}\end{aligned} \qquad (10b)$$

$Q_\tau$ is a factor that represents the amplification of the cumulative effects of selection over many generations on the genetic variance in fitness (Robertson 1961; Santiago and Caballero 2016).

As pointed out by Santiago and Caballero (2016), Equation (4) is derived from the asymptotic value of $N_e'(\tau)$ as time tends to infinity, reflecting the fact that the rate of fixations of mutations under selection is affected by the cumulative effects of the fitness variance on the rate of drift over a long period of time, whereas the effects on diversity are on a much shorter time-scale.

If the population size is large, time can be treated as a continuous variable. It is convenient to rescale time by dividing time in generations by $N_e = BN_0$, where $B$ is determined by Equation (7). The probability of no coalescence by time $T$ on this time-scale is:

$$\begin{aligned}P_{nc}(T) &\approx \exp\left\{-\int_0^T\frac{1}{N'(X)}\,dX\right\} \\ &= \exp\left\{-B\exp\frac{V_a^3}{V_m^2}\int_0^T\left[1 - \exp\left(-\frac{N_eV_mX}{V_a}\right)\right]\,dX\right.\end{aligned} \qquad (11a)$$

where $X = \tau/N_e$ and:

$$\frac{N_e V_m}{V_a} = \frac{\gamma[1 + \kappa \exp(-\gamma)]}{2[1 - \exp(-\gamma)]} \qquad (11b)$$

$$\frac{V_a^3}{V_m^2} = \frac{\alpha_0 [1 - \exp(-\gamma)]^3}{\gamma[1 + \kappa \exp(-\gamma)]^3} \qquad (11c)$$

The mean coalescence time relative to the fixation $N_e$ is given by:

$$\frac{B'}{B} = \int_0^\infty \frac{1}{N'(T)} T P_{nc}(T) \, dT = \int_0^\infty P_{nc}(T) \, dT \qquad (12)$$

If the value of $N_e$ has been found from Equation (7), numerical evaluation of Equation (12) allows the effect of HRI on neutral diversity to be determined.

## The expected neutral SFS

To determine the expected SFS at neutral sites in a sample of $k$ alleles, it is necessary in general to obtain an expression for the expected size of each successive interval between coalescent events

**Table 1.** Comparisons of simulation results for B and B′ in Supplementary Fig. 4 of Devi et al. (2023) with the predictions of Equations (7) and (12).

| L | $N_0 s$ | Simulation B | Predicted B | Simulation B′ | Predicted B′ |
|---|---|---|---|---|---|
| $10^3$ | 0.02 | 1.000 | 1.000 | 1.000 | 0.978 |
| $10^4$ | 0.02 | 1.000 | 0.996 | 1.000 | 0.976 |
| $10^5$ | 0.02 | 0.900 | 0.966 | 1.000 | 0.960 |
| $10^6$ | 0.02 | 0.700 | 0.815 | 0.800 | 0.873 |
| $10^3$ | 2 | 0.410 | 0.483 | 0.620 | 0.646 |
| $10^4$ | 2 | 0.250 | 0.285 | 0.400 | 0.438 |
| $10^5$ | 2 | 0.150 | 0.158 | 0.230 | 0.270 |
| $10^6$ | 2 | 0.080 | 0.081 | 0.140 | 0.155 |
| $10^3$ | 200 | – | 0.741 | 0.750 | $0.741^a$ |
| $10^4$ | 200 | 0.030 | 0.050 | 0.060 | 0.070 |
| $10^5$ | 200 | 0.010 | 0.009 | 0.020 | 0.021 |
| $10^6$ | 200 | 0.000 | 0.004 | 0.009 | 0.011 |

The simulations assumed $N_0 = 10^6$, $u = 3 \times 10^{-8}$, $\kappa = 3$.
[a]This value was obtained from Equation (6), which is more accurate than numerical integration of Equation (11a) in this region of parameter space.

in the gene tree for the nonrecombining block, as the numbers of nodes in the gene tree decrease from $k$ to 1 (Griffiths and Tavaré 1998). It is important to note that the gene tree applies to both selected and neutral sites, but the genetic diversity statistics considered here apply only to neutral mutations that obey the infinite sites model. The representation of a coalescent process with a temporally varying effective population size can be used for this purpose (e.g. Griffiths and Tavaré 1998, Polanski et al. 2002; Polanski and Kimmel 2003; O'Fallon et al. 2010; Walczak et al. 2012; Nicolaisen and Desai 2013; Strütt et al. 2025).

Here we use Equations (10), together with the results in section 2 of the Appendix, to calculate the trajectory of the effective population size at a time $\tau$ back in the past, $N_e'(\tau)$, and then substitute the results into Equation (52) of Polanski et al. (2002) for the expected SFS (see also Polanksi and Kimmel 2003). An alternative approach is to use the results of the coalescent process directly, as described in sections 1 and 2 of Supplementary File 1. This was, however, found to give substantially less accurate results than the alternative method (see Table 2). Both procedures assume that members of a set of ancestral alleles present at a given time in the past are exchangeable (O'Fallon et al. 2010); this assumption is violated if selection is acting on a block of linked sites, even if only neutral mutations are being studied, so that the results are likely to be only approximate.

A useful property of the SFS is that deviations from neutral expectation can be determined solely from the properties of segregating sites in a sample. As shown in section 2 of the Appendix, these properties can be used to obtain the measure of distortion $\Delta\theta_w$ proposed by Becher et al. (2020). This involves the ratio of the mean pairwise nucleotide diversity, $\pi$, to Watterson's $\theta_w$, which is given by the mean number of segregating sites in a sample of $k$ alleles ($S_k$) divided by the sum $a_k$ of the harmonic series $1/i$ from $i = 1$ to $k—1$ (Watterson 1975). It is defined as:

$$\Delta\theta_w = 1 - \frac{\pi}{\theta_w} \qquad (13)$$

The number of segregating sites used to calculate $\theta_w$ is insensitive to variant frequencies, in contrast to $\pi$. Since the expected values of $\pi$ and $\theta_w$ for neutral variants are equal at mutation-drift equilibrium with a constant $N_e$ (Watterson 1975, their ratio is expected to

**Table 2.** Comparisons of theoretical predictions of population statistics with the new simulation results.

| Parameters | $\pi_{neut}$ (sim) | $\pi_{neut}$ (pred) | $\pi_{sel}$ (sim) | $\pi_{sel}$ (pred.u) | $\pi_{sel}$ (pred.c) | $\pi_{sel}/\pi_{neu}$ (sim) | $\pi_{sel}/\pi_{neu}$ (pred.u) | $\pi_{sel}/\pi_{neu}$ (pred.c) | $\Delta\theta_w'$ (sim) | $\Delta\theta_w'$ (PK) | $\Delta\theta_w'$ (Coal) |
|---|---|---|---|---|---|---|---|---|---|---|---|
| No HRI, $\gamma_0 = 2$ | 0.02 | 0.02 | 0.0153 | 0.0153 | 0.0153 | 0.382 | 0.382 | 0.382 | 0 | 0 | 0 |
| No HRI $\gamma_0 = 20$ | 0.02 | 0.02 | 0.0020 | 0.0020 | 0.0020 | 0.050 | 0.050 | 0.050 | 0 | 0 | 0 |
| $L = 2500$ $\gamma_0 = 2$ | 0.0145 | 0.0120 | 0.0118 | 0.0069 | 0.0101 | 0.822 | 0.575 | 0.841 | $0.173 \pm 0.038$ | 0.163 | 0.122 |
| $L = 2500$ $\gamma_0 = 20$ | 0.0064 | 0.0055 | 0.0024 | 0.0018 | 0.0024 | 0.381 | 0.327 | 0.436 | $0.391 \pm 0.030$ | 0.344 | 0.182 |
| $L = 10,000$ $\gamma_0 = 2$ | 0.0104 | 0.0094 | 0.0091 | 0.0048 | 0.0075 | 0.873 | 0.511 | 0.798 | $0.290 \pm 0.033$ | 0.234 | 0.128 |
| $L = 10,000$ $\gamma_0 = 20$ | 0.0038 | 0.0035 | 0.0024 | 0.0013 | 0.0022 | 0.612 | 0.372 | 0.629 | $0.510 \pm 0.028$ | 0.455 | 0.233 |
| $L = 40,000$ $\gamma_0 = 2$ | 0.0080 | 0.0067 | 0.0072 | 0.0033 | 0.0060 | 0.900 | 0.492 | 0.896 | $0.357 \pm 0.029$ | 0.308 | 0.186 |
| $L = 40,000$ $\gamma_0 = 20$ | 0.0024 | 0.0024 | 0.0019 | 0.0009 | 0.0017 | 0.800 | 0.375 | 0.708 | $0.573 \pm 0.021$ | 0.533 | 0.651 |

The terms (sim) and (pred) refer to the simulated and theoretical values of the statistics in question; (pred.u) and (pred.c) refer to the uncorrected theoretical values of $\pi_{sel}$ and the theoretical values of $\pi_{sel}$ corrected for LD by the method described in the main text, respectively. The $\Delta\theta_w'$ column gives the mean values of the ratio of $\Delta\theta_w$ for neutral sites relative to its maximal value for a sample size of 20. PF and Coal refer to the Polanski et al. and coalescent SFS predictions, respectively. 95% confidence intervals of the mean are shown for $\Delta\theta_w'$; the other statistics have negligible CIs. A population of size 1,000 with a mutation rate of $10^{-5}$ and no mutational bias was simulated; the values with no HRI were obtained from the relevant theory.

be <1 if there is an excess of rare variants compared with this null expectation (Tajima 1989). If the SFS shows a higher number of low frequency variants than expected, as expected with HRI, the maximum value of $\Delta\theta_w$ ($\Delta\theta_{wm}$) can easily be found (Equation A.2c). This suggests that we can normalize $\Delta\theta_w$ by $\Delta\theta_{wm}$ to obtain a measure of distortion of the SFS toward low frequency variants that is less sensitive to the sample size than $\Delta\theta_w$ itself. The resulting statistic is identical to the normalized measure of Tajima's $D$ statistic proposed by Schaeffer (2002), except for a difference in sign. We have:

$$\Delta\theta'_w = \frac{\Delta\theta_w}{\Delta\theta_{wm}} \tag{14}$$

## Computer simulations

We ran forwards-in-time simulations using SLiM 4.1 (Haller and Messer 2023) in Wright-Fisher mode. SLiM is designed for diploids. To model haploid individuals, we removed any mutation from each individual's second genome in each generation. Our deleterious mutation model, detailed above, allows back mutations. To allow for this in the simulations, we created 2 mutation types: a "bookkeeping" variant to track mutation events at deleterious sites, and another representing the actual deleterious variant. Whenever a mutation happened at a deleterious site, a bookkeeping variant was placed on that site. If there was no deleterious variant present at that site and chromosome, the bookkeeping variant was replaced by a deleterious one. If there was already a deleterious variant at the site, both the deleterious and the bookkeeping variants were removed. This implies that $\kappa = 1$. We included a third mutation type to represent neutral variants, for which we did not implement back mutations. We set the global mutation rate to 2 times $u$, implying an equal chance that a mutation resulted in a bookkeeping or a neutral variant. This allowed us to analyze deleterious and neutral genotype matrices separately with a mutation rate of $u$ for each. We did not divide the genome into segments; deleterious and neutral mutations happened in an interspersed fashion, with neutral variants allowed to stack onto deleterious ones. We ran all simulations for 200,000 generations, since tracking variant frequencies over time indicated that simulations had essentially equilibrated by that time. At the end of a simulation run, we exported genotype matrices and variant site positions separately for neutral and deleterious variants. We ran simulations for 6 parameter sets: genome lengths $L \in \{2500, 10,000, 40,000\}$, $s$ values $\in \{10^{-3}, 10^{-2}\}$, $\kappa = 1$, $u = 10^{-5}$, $N = 1000$, with 200 replicate runs for each set.

## Semianalytical SFS statistics

To obtain the expected SFS for neutral sites under HRI with a given set of parameters, we used the mathematical machinery developed by Polanski and Kimmel (2003) and Polanksi et al. (2003), as described above. We numerically solved Equation (7) to obtain $B$. We then used Equations (10) to obtain the apparent population size trajectory due to HRI and BGS—a monotonically decreasing function of the number of generations back in time, $t$, which asymptotes as $t$ goes to infinity. For computational reasons, we discretized this function into 10 equal steps of population size decrease before using it as the input for computing the expected conditional SFS—see directory "PKsfs" in the Zenodo repository associated with this article.

## Results
### Levels of adaptation and neutral diversity under HRI
#### Fits to previous simulation results

Figures 1 and 2 compare $\bar{p}$ values from simulations, taken from Supplementary Fig. 3 of Devi et al. (2023), with the predictions of Equations (7) and (9), for the case of $\kappa = 1$ and $\kappa = 10$, respectively. The values of $\bar{p}$ from Equation (1) in the absence of HRI are also shown, providing a sense of the effect of HRI on the level of adaptation. The detailed results on which this section is based are shown in Supplementary File 2. In addition, Section 3 of Supplementary File 1 presents some analytical results on the effects on $B$ of changes in the net deleterious mutation rate $U$, the selection coefficient $s$ and the baseline effective population size $N_0$, which are in broad agreement with the observed patterns.

The 2 approximate formulae both provide fairly good fits to the simulation results when there is no mutational bias, although both predictions deviate somewhat from the dashed line that corresponds to a perfect fit (Fig. 1). In this case, Equations (9) perform worse than Equations (7) with $L = 10^4$, weak selection and a low mutation rate (upper left panel), underestimating the effects of HRI. With a strong mutational bias and $L = 10^4$ (Fig. 2), Equations (9) perform much worse, for both sets of selection and mutation parameters. With $L = 10^6$, both approximations give similar results, although Equations (9) fit somewhat better in the absence of mutational bias. Further comparisons, using the simulation results of McVean and Charlesworth (2000) and Comeron and Kreitman (2002) are shown in sections 4 and 5 of File S1.

We also investigated the extent to which the reduction in neutral diversity at sites embedded within a nonrecombining block of selected sites can be predicted by our approach. Table 1 compares the predictions of $B$ and $B'$ from the above equations with the simulation results of Devi et al. (2023), and Supplementary Table 1 in section 4 of Supplementary File S1 gives similar comparisons with the simulations of Comeron and Kreitman (2002). Overall, despite the approximations involved, agreement is satisfactory over a wide range of numbers of selected sites, strengths of selection, and mutational bias levels.

#### Fits to new simulation results

We used computer simulations to further investigate the effects of HRI on levels of adaptation and neutral diversity, with a special emphasis on the extent to which HRI affects the SFS at neutral sites. We also used them to determine whether the simulated effects of LD can be used to correct the diversity at selected sites and genetic variance at selected sites that is predicted by Equations (2)—see the subsection above on *The effect of HRI on the level of adaptation*, and section 1 of the Appendix. The detailed simulation results are shown in Supplementary File S3.

A summary of the main simulation results and predicted values for the statistics $\bar{p}$, $B$, and $B'$ is shown in Fig. 3. Table 2 shows the results for the neutral diversities $\pi_{neut}$, the simulated and uncorrected predicted values of the diversities at selected sites ($\pi_{sel}$), their predicted diversities corrected for the effects of LD using the LD estimates from the simulations ($\pi_{sel1}$), the $\pi_{sel}/\pi_{neut}$ ratios obtained from the simulations and the corresponding ratios obtained from the uncorrected and corrected formulae for $\pi_{sel}$, and $\Delta\theta'_w$.

The corrections for the effects of LD on diversity were obtained from Equation (A1a) of the Appendix, using the fact that all sites experience the same evolutionary forces. The sum of the $D$ values over all $L(L-1)/2$ pairs of sites, $\sum_{ij} D_{ij}$, was obtained for each replicate simulation, where $D = 0$ for any case in which one member of

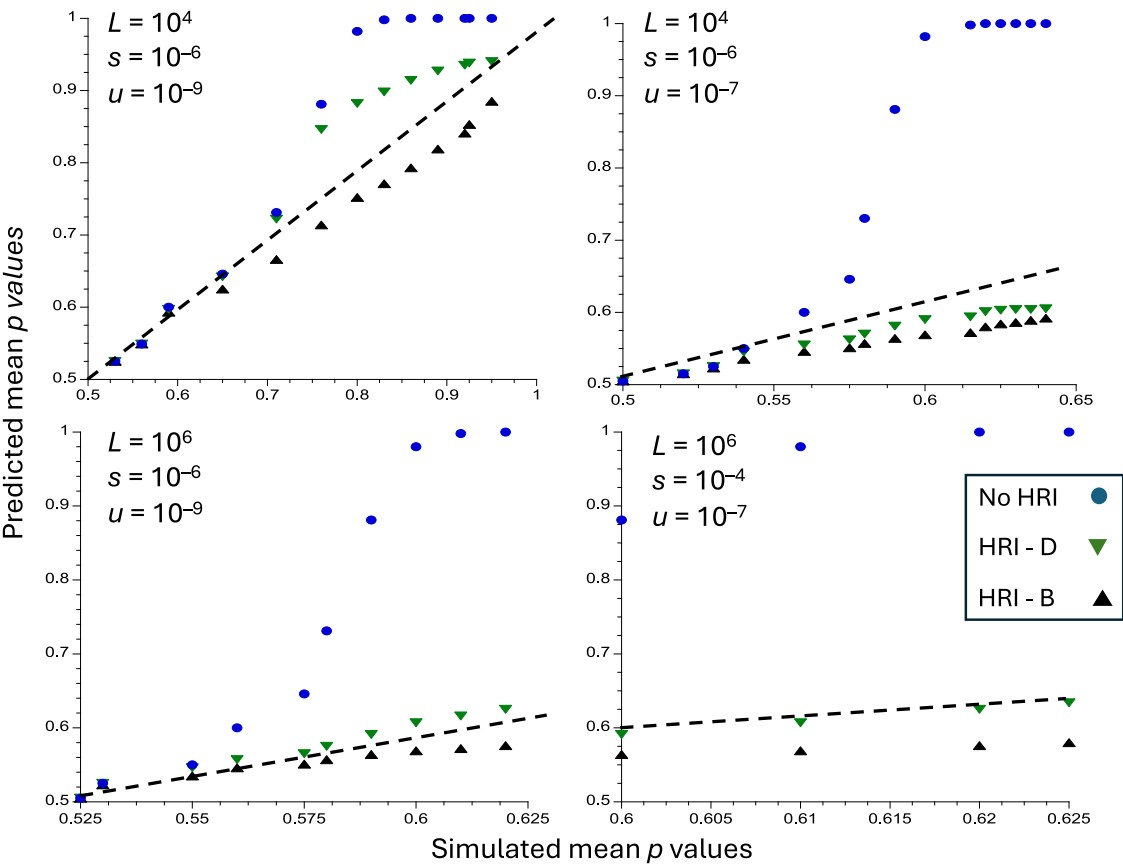

**Fig. 1.** Predicted vs simulated values of $\bar{p}$, the equilibrium mean frequency of the selectively favorable allele, for the case of no mutational bias ($k = 1$). The x axes show values of obtained from the simulation results shown in Supplementary Fig. 3 of Devi *et al.* (2023). The y axes correspond to the predicted values of the values of $N_0s$ used in the simulations. The upper panels are for $L = 10^4$ and the lower panels are for $L = 10^6$. The circles (blue) are the values of in the absence of HRI, obtained from Equation (1a) (labeled as No HRI); the triangles are obtained from Equations (1) and (7) (labeled as HRI—B) the inverted triangles (green) are the predicted values of in the presence of HRI obtained from Equations (9) (labeled as HRI-D). The dashed straight lines correspond to $y = x$; if the predicted values of p-bar with HRI were identical to the simulated values, they would fall on these lines.

a pair is not segregating. The correction for the mean diversity at a selected site, $C$, was obtained from the mean of the $D_{ij}$ over all pairs of selected sites ($\bar{D}$) as follows:

$$C = -2(L - 1)\bar{D} = -\frac{4}{L}\sum_{ij} D_{ij} \qquad (15)$$

The factor of $L-1$ arises because a given site potentially experiences the effects of LD at $L-1$ other sites. $C$ is then added to $\pi_{sel}$ in Equation (2a) to obtain the corrected value of the diversity at selected sites.

The simulations show that $\bar{D}$ is always negative, as expected from previous work on HRI with this type of model (e.g. McVean and Charlesworth 2000), so that the correction is invariably positive. Table 2 shows that this correction brings the predicted diversities at selected sites close to the simulated values, validating the approximations used to derive Equations (1). Figure 3 also confirms that $\bar{p}$, $B$, $B'$ are reasonably well predicted by the formulae derived here, as was previously seen in Figs. 1 and 2 and Table 1.

The behavior of $\Delta\theta'_w$, which measures the distortion of the gene genealogy toward longer external branches (and hence the distortion of the SFS toward an excess of rare derived variants) is perhaps of more interest, as the values of this statistic have not previously been predicted. Table 2 shows that, as expected, the value of $\Delta\theta'_w$ for a sample of 20 genomes increases as the effect of HRI on $B'$ increases. The value of $\Delta\theta'_w$ using the predictions based on the trajectory of $N_e$ backwards over time (Equations 10), combined with the equations of Polanski and Kimmel (2003)

that describe the corresponding SFS (see the subsection *Semianalytical SFS statistics*), provides an excellent fit to the simulation results. In contrast, the predictions based on the coalescent process formulae (Equations A8 and A9) substantially underestimate HRI effects. Figure 4 shows the simulated and predicted mean SFSs for a sample size of 20, showing the accuracy of the Polanski-Kimmel predictions; comparisons with the SFS expected in the absence of HRI illustrate the considerable distortion toward low frequency variants at neutral sites caused by HRI for most of the frequency range.

Supplementary Fig. 1 in section 6 of Supplementary File 1 summarizes the distributions of the summary statistics used here across the 200 replicates simulations of each parameter set. A dataset on a single nonrecombining genomic region should correspond to a single outcome of the evolutionary process, so that the boxes and whiskers in these plots provide a picture of the probable spread of possible observations for a given parameter set. $\Delta\theta'_w$ is the noisest of the statistics, followed by $B'$ when selection is weak. In other cases, the mean values of the statistics provide a fairly good idea of what to expect for a single observation.

## Discussion

### Effects of HRI on the level of adaptation

The results presented here confirm the conclusion from several earlier studies that HRI induced by blocks of tightly linked sites under purifying selection can have major effects on the level of

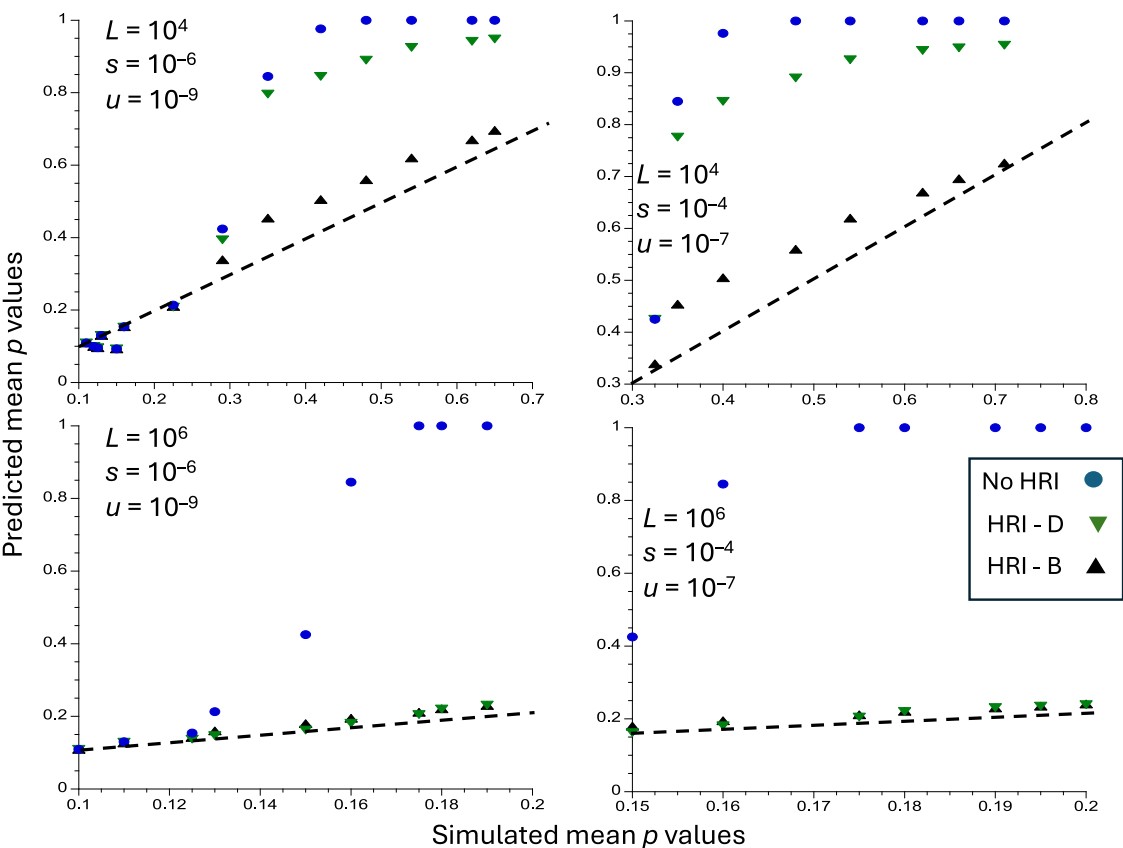

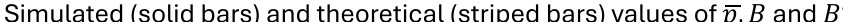

**Fig. 2.** Predicted vs simulated values of $\bar{p}$, the equilibrium mean frequency of the selectively favorable allele, for the case of no strong bias ($\kappa = 10$). The x axes show values of obtained from the simulation results shown in Supplementary Fig. 3 of Devi *et al.* (2023). The y axes correspond to the predicted values of $\bar{p}$, for the values of $N_0 s$ used in the simulations. The upper panels are for $L = 10^4$ and the lower panels are for $L = 10^6$. The circles (blue) are the values of $\bar{p}$, in the absence of HRI, obtained from Equation (1a) (labeled as No HRI); the triangles are obtained from Equations (1) and (7) (labeled as HRI—B) the inverted triangles are the predicted values of in the presence of HRI obtained from Equations (9) (labeled as HRI-D). The dashed straight lines correspond to $y = x$; if the predicted values of $\bar{p}$, with HRI were identical to the simulated values, they would fall on these lines.

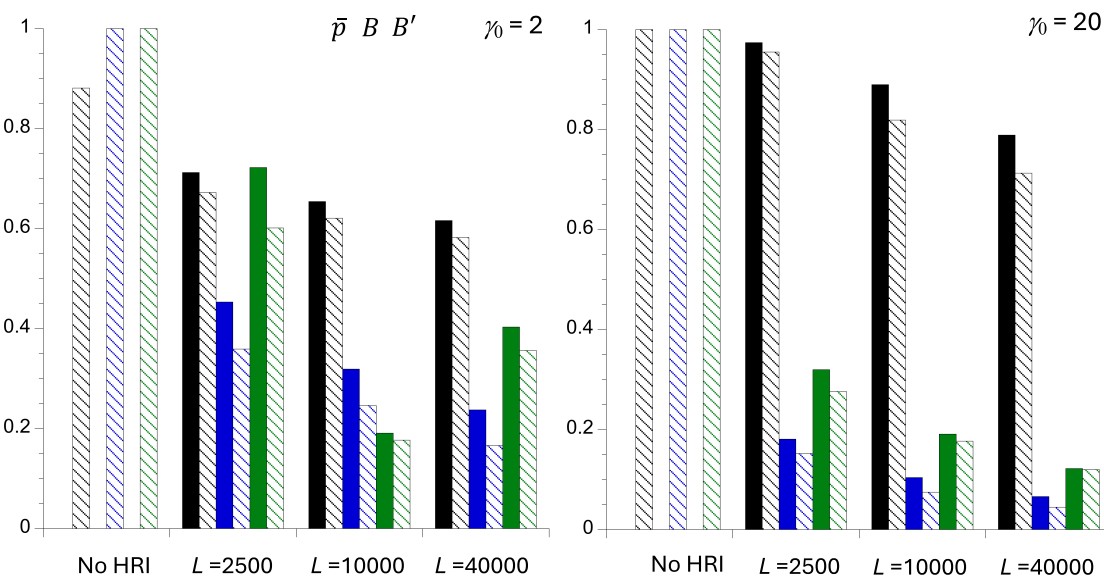

**Fig. 3.** The mean simulated values (solid bars) and theoretical values (striped bars) of the equilibrium mean frequency of the favored allele ($\bar{p}$), the ratio of the fixation $n_e$ to $N_0$ ($B$), and the ratio of the coalescence $n_e$ to $N_0$ ($B'$), for the cases of no HRI and 3 different numbers of selected sites ($L$). The bars representing $\bar{p}$, $B$, and $B'$ are the left-hand, middle, and right-hand bars for each case. The left-hand panel is for $\gamma_0 = 2$ and the right-hand panel is for $\gamma_0 = 20$. $N_0 = 1000$, $u = 10^{-5}$, and $\kappa = 1$ (i.e, no mutational bias).

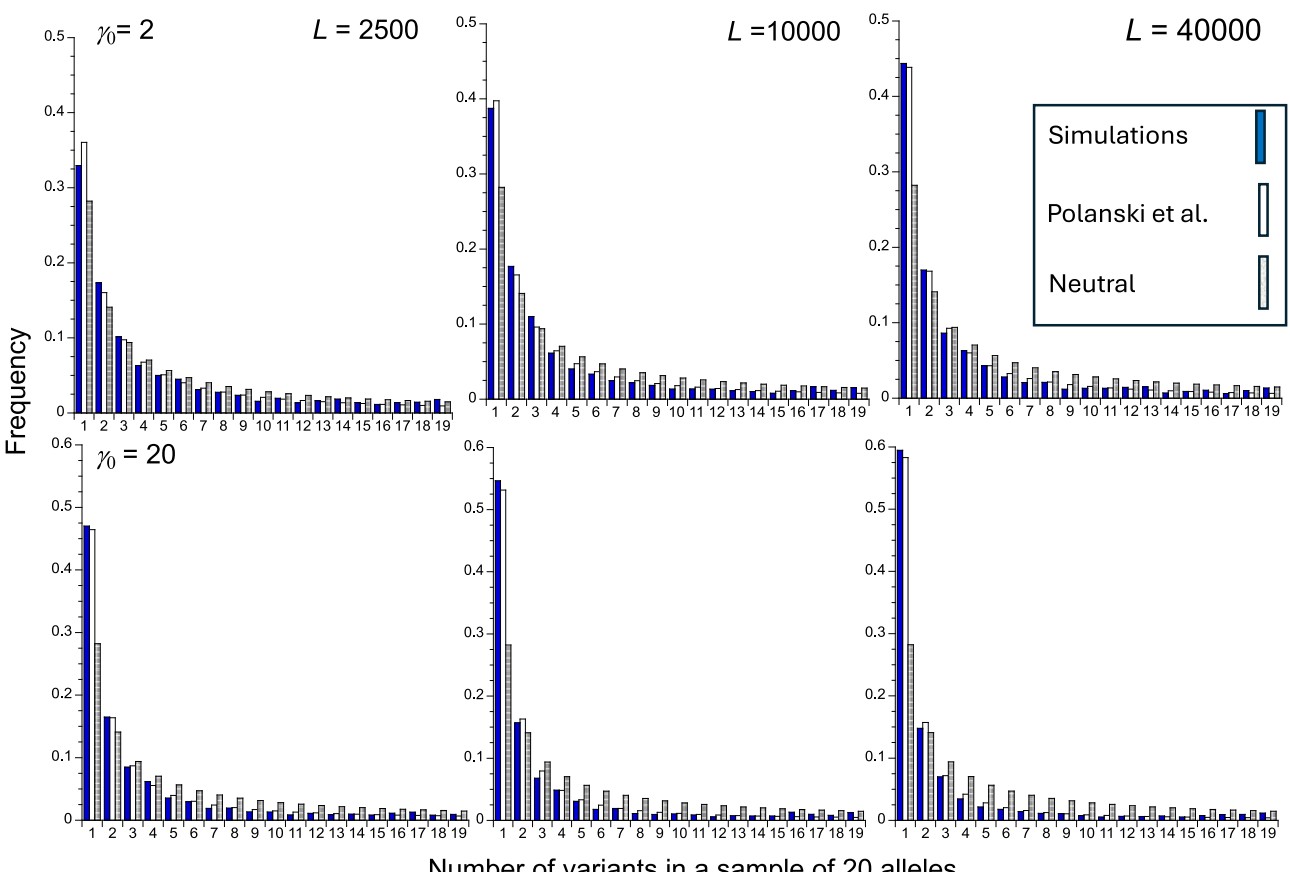

**Fig. 4.** Site frequency spectra for a sample of 20 neutral alleles in a haploid, nonrecombining genome, with no mutational bias, for the sets of parameters used in Fig. 3. For a given number of variants (1, 2, etc.) the bars from left to right are as follows: the SFS for the pooled simulation results (filled bar), the Polanski *et al.* (2023) prediction (empty bar), and the SFS in the absence of HRI (stippled bar). The top panel is for $\gamma_0 = 2$ and the bottom panel is for $\gamma_0 = 20$. The left-hand, middle, and right-hand panels are for $L = 2500$, 10,000, and 40,000, respectively.

adaptation as measured by the mean frequency ($\bar{p}$) at which selected sites are occupied by favorable alleles in a genome (Li 1987; Comeron *et al.* 1999; McVean and Charlesworth 2000; Tachida 2000; Comeron and Kreitman 2002; Charlesworth *et al.* 2010; Devi *et al.* 2023). As described in the section *Results: levels of adaptation and neutral diversity under HRI*, Equations (1), (7), and (8) provide a simple and rapid method for obtaining an approximate expression for the magnitude of this effect. In the present paper, the variable $B$ measures the factor by which the baseline effective population size in the absence of HRI ($N_0$) must be multiplied in order to obtain the $N_e$ for use in the standard diffusion equation formula for the fixation probability of a new mutation (Kimura 1964; Santiago and Caballero 2016; Buffalo and Kern 2024)—the "fixation $N_e$" of Devi *et al.* (2023). These effects do not require a small population size; rather, a combination of the number of selected sites, the mutation rate and $N_0s$ determines $B$ in a highly nonlinear fashion.

The $\bar{p}$ is related to $B$ via the Li-Bulmer equation (Equation 1a), where $\gamma = 2BN_0s = B\gamma_0$ in the presence of HRI, and provides a measure of codon usage bias under the simple model where $A_1$ at a site represents the preferred codon and $A_2$ represents an unpreferred alternative (Li 1987; Bulmer 1991; McVean and Charlesworth 1999). A large body of evidence suggests that $\gamma$ for synonymous site variants is usually of the order of one in normally recombining genomic regions of species where there is evidence for selection on codon usage, but with considerable variation between genes, associated with factors such as gene expression levels (Sharp

*et al.* 2005, 2010; Hershberg and Petrov 2008). As found in the simulation studies cited above, a sufficiently large number of such weakly selected sites can produce a strong reduction in $B$ and hence in $\bar{p}$, in the absence of any other types of selection, such as purifying selection at nonsynonymous sites. This is illustrated in Fig. 5 for the panels with $\gamma_0 = 1$ (for the detailed results, see Supplementary File 4).

This pattern is consistent with the evidence for reductions in $B$ and $B'$ in genomes and genomic regions where recombination is rare or absent, as reflected in levels of codon usage and silent site diversity, respectively (e.g. Qiu *et al.* 2011; Szövényi *et al.* 2011; Charlesworth and Campos 2014; Hough *et al.* 2014). This is especially true for neo-Y chromosomes in *Drosophila*, such as that of *D. miranda,* where a whole chromosome arm has stopped recombining due to the lack of recombination in males (e.g. Mahajan *et al.* 2018). For the *D. miranda* neo-Y chromosome, silent site diversity is approximately 1% of the genome-wide average (Bartolomé and Charlesworth 2006). Smaller nonrecombining regions in *Drosophila*, such as the centromeric regions, have silent site diversities of the order of 10% of the genome-wide average (Becher *et al.* 2020). Of course, other factors, such as selective sweeps of favorable mutations and HRI effects of strongly selected nonsynonymous mutations on weakly selected synonymous mutations are also likely to be involved in causing these patterns.

With $\gamma_0 = 1$ in the absence of HRI and a mutational bias of $\kappa = 2$ toward deleterious alleles, the ratio of $\bar{p}$ to its value of 0.576 in the absence of HRI declines almost linearly with the $\log_2$ of $L$, reaching

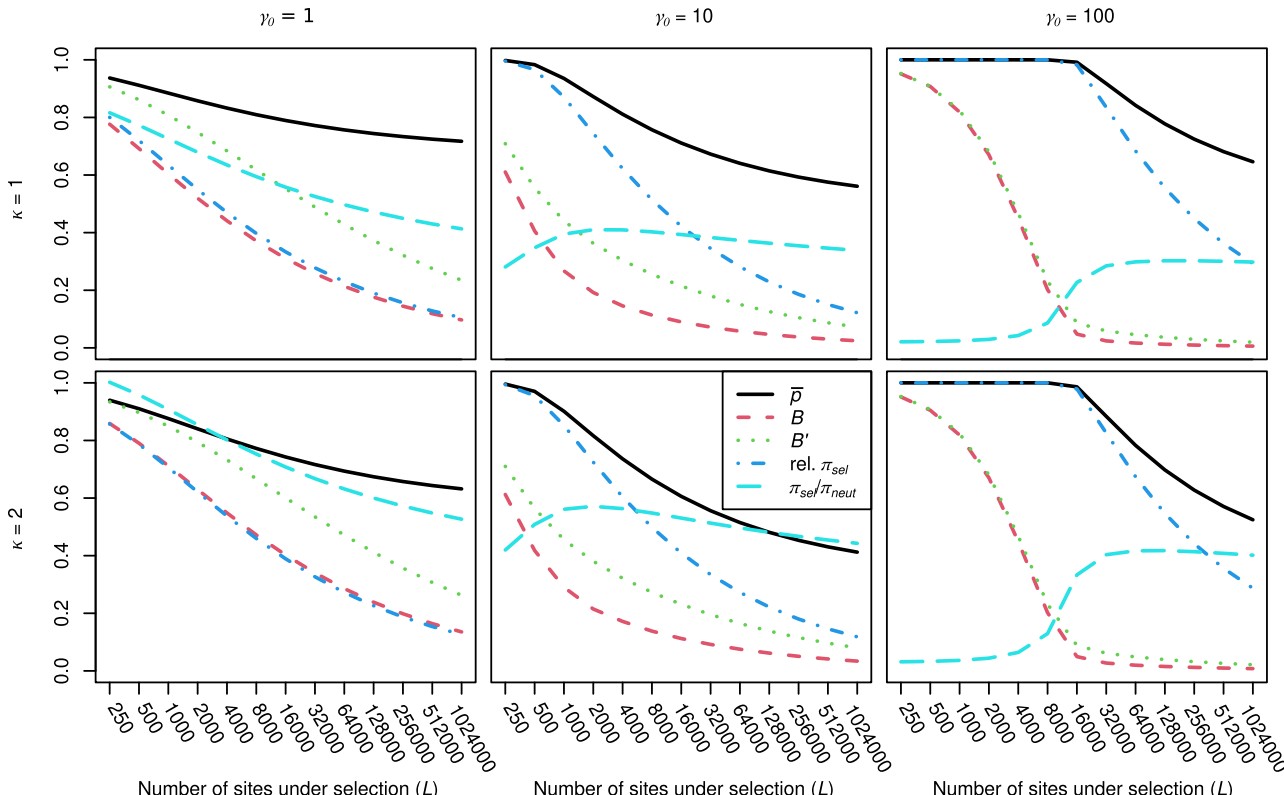

**Fig. 5.** Plots of the approximate values of population statistics obtained from equations (2a), (7), and (12) (Y axes) against the number of selected sites (X axes; log₂ scale) for mutational biases of $\kappa = 1$ (upper panels) and 2 (lower panels), with $\gamma_0 = 1$ (left-hand panels), $\gamma_0 = 10$ (middle panels), and $\gamma_0 = 100$ (right-hand panels). The ratio of $\pi_{sel}$ with HRI to its value in the absence of HRI is denoted by "Relative $\pi_{sel}$."

~0.65 with 1 million selected sites, corresponding to 2,000 genes with 500 synonymous sites. This number of genes is comparable with those in a typical *Drosophila* chromosome arm (Misra *et al.* 2002), and so would apply to a Drosophila neo-Y chromosome. The value of B itself declines much faster, with a value of ~0.1 with 1 million sites. The fixation probability of a deleterious mutation relative to that of a neutral mutation is equal to $\gamma/[\exp(\gamma)-1]$, and that of a favorable mutation is $\gamma/[1-\exp(-\gamma)]$ (Kimura 1964). These quantities take values of 0.58 and 1.58 with no HRI, respectively, becoming 0.95 and 1.05, respectively, with 1 million selected sites. This implies a sizeable increase in the rate of substitution of deleterious mutations, and a decrease for favorable ones, in a large nonrecombining genome. This effect can explain features such as the much faster rate of substitutions of mutations toward unpreferred codons relative to preferred codons in genes on the newly nonrecombining neo-Y chromosome of *Drosophila miranda* (Bartolomé and Charlesworth 2006), even in the absence of any other source of HRI effects.

With stronger selection and hence a larger value of $\gamma_0$, the probability of fixation of a favorable allele when B is small can still be relatively high, and the probability of fixation of its deleterious alternative correspondingly low, simply because $\gamma = B\gamma_0$ remains large. The lower middle panel of Fig. 5 shows that B with $\kappa = 2$ and $\gamma_0 = 10$ is ~0.05 with 1 million selected sites, whereas the ratio of $\bar{p}$ to its value in the absence of HRI is ~0.42. The fixation probabilities of deleterious and favorable mutations, relative to neutrality, change from values of 0.00045 and 10, respectively, in the absence of HRI to 0.98 and 1.03 with 1 million selected sites.

With $\gamma_0 = 100$, there is a similar pattern, except that $\bar{p}$ shows a threshold effect, with little or no decline with L until there are

~50,000 selected sites, corresponding to B reaching a value of ~0.05, so that $\gamma = 5$. This reflects the well-known quasi-threshold behavior of $\bar{p}$ in Equation (1a) as a function of $\gamma$. With strongly selected variants, such as nonsynonymous mutations, it thus may be hard to detect reduced adaptation in nonrecombining genomic regions simply from the abundance of putatively deleterious mutations, unless the region includes hundreds of thousands of selected sites. Nonetheless, increased ratios of nonsynonymous to synonymous substitutions have been detected in nonrecombining genomic regions such as the *D. miranda* neo-Y chromosome (Bartolomé and Charlesworth 2006), the Y chromosome of *Rumex hastulus* (Hough *et al.* 2014) and the pericentromeric and telomeric regions of *D. melanogaster* and its relatives (Campos *et al.* 2012, 2014).

It is important to remember that, in the absence of recombination, B can depart massively from the classical BGS prediction of Equation (6); the same applies to the measure B′, which we have used to predict the effects of HRI on linked neutral sites (Equation 12). The classical formula assumes that all selected sites are at deterministic mutation-selection equilibrium and has been used either to predict neutral diversity at linked sites (Charlesworth *et al.* 1993), and the fixation probabilities of mutations that are under much weaker selection than the selected ones (Charlesworth 1994; Peck 1994; Orr and Kim 1998; Johnson and Barton 2002). Departures from the predictions of Equation (6) can occur in the absence of reverse mutations, even with $\gamma_0 \gg 1$, provided that U/s is so large that the expected number of mutation-free individuals in the population, $N_0 \exp(-U/s)$, is small enough for them to be vulnerable to stochastic loss, so that Muller's ratchet operates in the absence of reverse mutations

from mutant to wild-type (Charlesworth *et al.* 1993, 1995; Gordo *et al.* 2002; Nicolaisen and Desai 2013; Good *et al.* 2014; Cvijovic *et al.* 2018, Melissa *et al.* 2022). If $\gamma$ is of order 1 or less, deleterious allele frequencies may depart considerably from their deterministic equilibria, which again means that Equation (6) is inappropriate. Equations (7), (8), and (12) thus provide a more robust means than Equation (6) of predicting the consequences of selection against deleterious mutations in a low recombination environment.

## Effects of HRI on the ratio of nucleotide diversities at selected sites vs neutral sites

Another method for detecting a weakening of the efficacy of selection caused by HRI has used the ratio of mean nucleotide site diversity at selected sites ($\pi_{sel}$) to mean diversity at putatively neutral sites in the same genomic region ($\pi_{neut}$). In analyzing data, this approach commonly uses the ratio of the mean nonsynonymous site diversity ($\pi_N$) to that for synonymous sites ($\pi_S$), and in related measures, such as the diversity ratio at 0-fold coding sites ($\pi_0$) to that for 4-fold degenerate sites ($\pi_4$) (e.g. Charlesworth and Campos 2014; Bast *et al.* 2018; Castellano *et al.* 2018). Our results show that $\pi_{neut}$ for a set of neutral sites embedded in a nonrecombining section of the genome is well approximated by Equations (7) and (12), which determine $B'$, the ratio of $\pi_{neut}$ to its value in the absence of HRI. $\pi_{neut}$ is given by the limit of Equation (2a) as $\gamma$ approaches zero, $4N_0 u/(1 + \kappa)$, assuming that neutral sites are at statistical equilibrium with respect to their base composition and $4N_0 u \ll 1$.

A complication, however, is that the simple use of $2BN_0s$ for $\gamma$ in the equation for the diversity at selected sites (Equation 2a) underestimates $\pi_{sel}$, since the effect of LD in weakening selection against deleterious alleles is not fully taken into account simply by using $2BN_0s$ in this equation (see section 1 of the Appendix). While this effect can be approximately corrected for if the sum of the $D$ values across all pairs of sites is known (see section 1 of the Appendix and Table 2), this sum is not currently predictable analytically, except when selection is sufficiently strong that mutations are kept close to their frequencies under mutation-selection balance (Roze 2021).

The extent to which $\pi_{sel}/\pi_{neut}$ is underestimated can be assessed using the results in Table 2. With no HRI, $\pi_{sel}/\pi_{neut}$ is 0.380 for $\gamma_0 = 2$ and 0.050 with $\gamma_0 = 20$. With $L = 2500$, the simulated and uncorrected values of $\pi_{sel}/\pi_{neut}$ are 0.875 and 0.511 for $\gamma = 2$; for $\gamma_0 = 20$, the corresponding values are 0.375 and 0.327. With $L = 10000$, the simulated and uncorrected values of $\pi_{sel}/\pi_{neut}$ are 0.875 and 0.510 for $\gamma_0 = 2$; for $\gamma_0 = 20$, the corresponding values are 0.632 and 0.371. The extent of underestimation of $\pi_{sel}/\pi_{neut}$ without correcting for LD clearly gets worse with stronger selection and larger numbers of selected sites within the nonrecombining region. Nevertheless, for the parameters used in Table 2, Equations (2a) and (7) still predict a considerable increase in $\pi_{sel}/\pi_{neut}$ due to HRI.

Predictions based on these equations are shown in Fig. 5 for a much wider range of values of $L$ and selection parameters (the detailed results are presented in section 4 of Supplementary File 1). Somewhat unexpectedly, $\pi_{sel}/\pi_{neut}$ decreases with $L$ when selection is very weak ($\gamma_0 = 1$), decreasing almost linearly from a value of 0.92 and 1.09 with $\kappa = 1$ and 2, respectively, to ~0.42 and 0.58 at 1 million sites. This reflects the much slower decrease in $B'$ with $L$ compared with $B$, so that neutral diversity declines more slowly that the effective strength of selection against deleterious mutations. However, with stronger selection, $B'$ and $B$ are much closer, and $\pi_{sel}/\pi_{neut}$ increases substantially as $L$ increases, either reaching a plateau ($\kappa = 2$) or declining slightly ($\kappa = 1$). Given the inaccuracies

in the formula for $\pi_{sel}$, the exact values of $\pi_{sel}/\pi_{neut}$ should be treated with caution, but the general patterns are probably approximately correct. The overall magnitude of the effect of HRI in causing increased values of $\pi_{sel}/\pi_{neut}$ is underestimated by these results, so they can be regarded as conservative predictions of this measure of the effect of reduced efficacy of selection.

It might be thought that using $B'$ instead of $B$ in Equation (2a) would yield better predictions of $\pi_{sel}/\pi_{neut}$, since $B'$ values are more closely related to the behavior of segregating variants. This is not, however, borne out by the results in Fig. 3. Applying the values of $B'$ and mean $\pi_{neut}$ obtained from the simulations to Equation (2a), the predicted values of $\pi_{sel}/\pi_{neut}$ are 0.853 and 0.311 for $L = 2,500$ with $\gamma_0 = 2$ and 20, respectively, and 0.904 and 0.504 with $L = 1,000$ with $\gamma_0 = 2$ and 20, respectively. Comparing these with the simulation values for $\pi_{sel}/\pi_{neut}$ presented above shows that $\pi_{sel}/\pi_{neut}$ is over-predicted when $\gamma_0 = 2$ and underpredicted when $\gamma_0 = 20$, with stronger underprediction than overprediction.

This result implies that it may be unsafe to use estimates of $N_e$ from levels of putatively neutral diversity to make quantitative inferences concerning the effects of HRI on sites under selection. As an example of such an inference, Castellano *et al.* (2018) used the relation between $\ln(\pi_N/\pi_S)$ and $\ln(\pi_S)$ across different regions of the genome of *D. melanogaster* that differ in the rate of recombination to infer the shape parameter of the distribution of mutational effects (DFE) of deleterious nonsynonymous mutations, on the assumption that this is a gamma distribution, and that the slope of this relationship is equal to the negative of the shape parameter of the gamma distribution (Welch *et al.* 2008). Their estimate of this parameter (~0.5) was much larger than the value of ~0.3 commonly found for *D. melanogaster* from population genomics methods (e.g. Kousathanas and Keightley 2013). The failure of $B'$ to accurately predict diversity at selected sites in the presence of HRI could be a source of this discrepancy.

Another caveat concerning the use of an increase in $\pi_{sel}/\pi_{neut}$ as evidence for a reduced efficacy of selection on nonsynonymous sites is that a fraction of such sites could be selectively neutral, and the rest are subject to such strong purifying selection that they are insensitive to HRI. In such a case, $\pi_{sel}/\pi_{neut}$ would automatically increase with a reduction in $\pi_{neut}$ (Campos *et al.* 2014). Other information is thus required to conclude definitively that an increase in $\pi_{sel}/\pi_{neut}$ in low recombination genomic regions is caused by HRI, as discussed by Campos *et al.* (2014). If a sufficient number of genes is available for comparing regions with differing recombination rates with good statistical accuracy, a useful criterion for the operation of HRI would be a lower value of mean $\pi_{sel}$ in regions with low recombination rates compared with recombining regions (see Figs. 3 and 5), provided that the 2 regions do not consistently differ in other properties affecting $\pi_{sel}$.

In contrast, the hypothesis that a fraction $P_s$ of selected sites is under strong selection and immune to HRI, and a fraction $(1—P_s)$ behaves neutrally, predicts an increase in $\pi_{sel}$ as $N_e$ decreases, due to HRI. With this model, $\pi_{sel} = P_s\pi_{strong} + (1 - P_s)B'\pi_{neut0}$, where $\pi_{strong}$ is the (constant) diversity at strongly selected sites and $\pi_{neut0}$ is the neutral diversity in the absence of HRI. The ratio of $\pi_{sel}$ to its value with $B' = 1$ is thus given by:

$$R(\pi_{sel}) = 1 - \frac{(1 - P_s)(1 - B')\pi_{neut0}}{P_s\pi_{strong} + (1 - P_s)\pi_{neut0}} \tag{16}$$

This ratio is an increasing function of $B'$, in contrast to what is expected when diversity at selected sites is affected by HRI, as described above. This test was applied by Campos *et al.* (2014),

whose Tables 1 and 2 showed that the mean value of $\pi_N$ over 225 genes in 3 autosomal regions of *D. melanogaster* genomes with no crossing over in a sample from a Rwandan population was 0.00069, compared with a mean value of 0.00143 over 7,099 genes in autosomal regions with a "normal" rate of crossing over, a ratio of 0.48, with a highly significant difference. In contrast, the corresponding mean values of $\pi_S$ for the 2 regions were 0.00154 and 0.0141, a ratio of 0.109; the values of $\pi_N/\pi_S$ for the 2 regions were 0.254 and 0.101, respectively. These observations are thus consistent with the hypothesis that the DFE for nonsynonymous mutations is such that HRI results in decreased $\pi_N$, rather than the increase predicted by Equation (16).

Of course, such a test is not conclusive, since other factors might influence the properties of the genes in the regions. Other criteria, including reduced frequencies of optimal codons, GC content, and an increase in $d_N/d_S$, and in the lengths of introns in low recombination regions in this example (Campos *et al.* 2012) all point to effects of HRI in undermining the efficacy of purifying selection in this case.

## Effects of HRI on diversity at neutral sites

As has been known since the earliest study of the effects of BGS on neutral diversity (Charlesworth *et al.* 1993), HRI causes departures from the deterministic equilibrium frequencies of deleterious mutations on which Equation (6), leading to a considerable weakening of its effect on diversity at linked neutral sites, as measured here by $B'$. Nevertheless, increased levels of HRI are expected to lead to reduced values of $B'$ (e.g. Gordo *et al.* 2002; Kaiser and Charlesworth 2009; Seger *et al.* 2010; Good *et al.* 2014; Santiago and Caballero 2016; Hough *et al.* 2017). This pattern is illustrated in Fig. 5, where $B'$ is plotted against the number of sites under selection (L), for several strengths of selection and 2 levels of mutational bias. There is a strong tendency for the rate of decrease in $B'$ to decline as L increases (note that L is measured on a $\log_2$ scale), although it does not seem that a true asymptote is reached, as was initially proposed by Kaiser and Charlesworth (2009) on the basis of simulations. With a million selected sites and a scaled selection coefficient of 100, neutral diversity is reduced to a very low level, ~2% of its value in the absence of HRI. This is consistent with the evidence for extreme reductions in levels of neutral or nearly neutral diversity in large nonrecombining genomic regions (reviewed by Charlesworth and Jensen 2021). Note, however, that Fig. 3 shows that $B'$ tends to be somewhat underestimated by the approximations used here, especially when L is small.

## Distortions of the SFS at neutral sites

As has been shown in many previous studies (Charlesworth *et al.* 1995; Santiago and Caballero 1998; O'Fallon *et al.* 2010; Nicolaisen and Desai 2013; Good *et al.* 2014; Cvijvocic *et al.* 2018), HRI is also expected to cause a distortion of the SFS at linked neutral sites toward more than expected low frequency derived variants in sequences in nonrecombining genome regions—see Table 2 and Fig. 4.

Many empirical studies have detected such an effect in comparisons of low and high recombination genomic regions (reviewed by Charlesworth and Jensen 2021). Tajima's D statistic (Tajima 1989) is a commonly used measure of the extent of the distortion of the SFSs, compared with the equilibrium neutral expectation—see Equation (S10) in section 1 of Supplementary File 1. This statistic has several undesirable properties, notably a dependence on the number of segregating sites in the sample, which in turn depends on the size of genome region analyzed, as well as on the sample size. Other proposed measures include Schaeffer's

$D_{min}$ statistic, which compares the estimated Tajima's D with its maximum absolute value for a given sample size and number of segregating sites (Schaeffer 2002), the $\Delta\pi$ statistic of Langley *et al.* (2014) and the $\Delta\theta_w$ statistic of Becher *et al.* (2020)—see Equation (13). Here, we used a modification of $\Delta\theta_w$, $\Delta\theta'_w$, which is the ratio of $\Delta\theta_w$ to its maximum possible value when there is an excess of rare variants (Equation 14); this is equivalent to—$D_{min}$. This statistic is not completely free of dependence on the sample size, but it is far less sensitive to sample size than Tajima's D, as can be seen from Supplementary Table 4 in section 7 of Supplementary File S1, which is based on the simulations used to generate Table 2. Moreover, section 2 of the Appendix shows that $\Delta\theta'_w$ can be estimated from the SFS alone, without reference to the number of segregating sites, and is therefore useful for analyzing population genomic data. It thus seems to have considerable utility as a measure of the skew of the SFS away from equilibrium neutral expectation.

The present approach does not, however, fully describe the processes involved in determining the SFS. Fixations of either $A_1$ or $A_2$ occur at rates governed by the products of the proportions of sites fixed for $A_2$ or $A_1$ ($\bar{p}$ and $1-\bar{p}$, respectively) and their expected rates of substitution. With L sites under selection, the overall expected rate of fixation of new mutations per generation is given by:

$$\lambda = \frac{2Lu\gamma}{[1+\kappa\exp(-\gamma)][\exp(\gamma)-1]} \tag{17}$$

where $\gamma = B\gamma_0$ (Charlesworth and Charlesworth 2000, p.275, Equation 6.11).

This quantity can become large when the number of sites is large, allowing "sweeps" of new mutations of either type to become frequent. For example, for the example in Table 2 with $N_0 = 1000$, $\gamma_0 = 20$, $u = 10^{-5}$, $\kappa = 1$, and $L = 40000$, the simulation results gave $B = 0.066$, so that $\gamma = B\gamma_0 = 1.32$, giving $\lambda = 0.40$, i.e. one sweep every 2.5 generations in the simulations (note, however, that if the parameters are rescaled to a population size of $10^6$, this corresponds to a sweep every 2,500 generations). There is, therefore, a chance that a derived neutral variant can be caught up in a sweeping haplotype and be present at a high frequency at the time the population is sampled, leading to a small fraction of sites with derived alleles at high frequencies (e.g. Fay and Wu 2000). This possibility is not captured by the approximations used here, which consider only the effect on the SFS of the apparent population expansion created by hitchhiking effects (Equations 10). Devi *et al.* (2023) used an argument based on models of clonal interference between sweeping mutations to obtain approximate expressions for the effects of HRI, such as Equations (9), but did not consider the SFS.

Cvijovic *et al.* (2018) analyzed the case of irreversible deleterious mutations for a nonrecombining genome with $Lu/s >> 1$, and derived approximations for the SFS that predict an uptick in the probabilities of high frequency derived variants at the extreme end of the distribution of frequencies. Careful examination of Fig. 4 shows that this effect is visible when the frequencies of the classes with 19 and 18 derived variants in a sample of 20 are compared with each other, with class 19 being consistently more frequent than 18, even with $\gamma = 2$ ($Lu/s = 2$). The effect is more noticeable with larger sample sizes (see Supplementary Table 5 in Supplementary File 1, and the frequency spectra in Supplementary File 3). This deficiency in the application of the Polanski *et al.* (2002) formula to the SFS under HRI probably explains why Fig. 4 shows that it consistently slightly

underestimates the probabilities of relatively high frequency derived variants. The overall effect is, however, small, and the $\Delta\theta'_w$ summary statistic is accurately predicted by this formula, at least for the parameter sets displayed in Table 2.

## Limitations of the study and future prospects

While our results provide some useful insights into the effects of HRI on patterns of genetic diversity and levels of adaptation when recombination is absent, they have some obvious limitations. First, only a single selection coefficient is assigned to each site under selection; in reality, a wide distribution of selection coefficients is likely to apply to deleterious mutations (e.g. Eyre-Walker and Keightley 2009; Kim et al. 2017). It is not clear how to deal with this problem, as previous simple examples involving 2 strengths of selection suggest that integrating Equations (10) over a distribution of selection coefficients (Santiago and Caballero 2016; Buffalo and Kern 2024) may not be adequate in low recombination regimes, a topic that needs to be explored in future simulation studies. Our concern is that sites under strong selection are likely to have larger effects on weakly selected sites than vice-versa (Gordo and Charlesworth 2001; Kaiser and Charlesworth 2010). Devi et al. (2023) have examined the properties of $\bar{p}$ with up to 3 classes of selected sites for the reversible mutation model by a heuristic argument, but a definitive treatment of this problem seems to be hard to obtain.

Second, there is evidence that gene conversion events occur with significant frequencies in noncrossover regions of genomes, at least in Drosophila (Comeron et al. 2012) and humans (Palsson et al. 2025), and that low frequency recombination events can happen in apparently asexual lineages such as the Bdelloid rotifers (Vakhrusheva et al. 2020; Laine et al. 2022). It is therefore of interest to examine the robustness of our conclusions to low frequency recombination events, although Kaiser and Charlesworth (2009) found only small effects of gene conversion in their simulations. Finally, we do not consider diploidy, where the recessivity or partial recessivity of deleterious mutations can result in effects of associative overdominance or pseudo-overdominance that act in the opposite direction to the HRI effects discussed here (Zhao and Charlesworth 2016; Becher et al. 2020; Gilbert et al. 2020; Abu-Awad and Waller 2023) Further developments of analytical approximations and comparisons with simulation results are needed to investigate these questions further.

## Data availability

No new data or reagents were generated for this work. The codes for the computer programs used to generate the results described above are available on https://doi.org/10.5281/zenodo.14024806. The numerical results used to produce the figures and tables are presented in Supplementary Files S1–S4.

Supplemental material available at GENETICS online.

## Acknowledgments

We thank Deborah Charlesworth, Denis Roze, Enrique Santiago and an anonymous reviewer for their comments on an earlier version of this paper. This work has made use of the resources provided by the Edinburgh Compute and Data Facility (ECDF) (http://www.ecdf.ed.ac.uk/).

## Funding

This work was not funded.

## Conflicts of interest

The authors declare no conflict of interest.

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

# Appendix

## 1. The effects of LD

For a haploid population, the total additive variance in fitness, $V_a$, is equal to $V_g$ plus twice the sum of terms involving the covariances in fitness between all pairs of sites (Bulmer 1980). For a pair of sites $i$ and $j$, the covariance is $C_{ij} = D_{ij}s^2$, where $D_{ij}$ is the coefficient of LD for this pair of sites, giving a contribution to $V_a$ of $2\,D_{ij}s^2$. In addition, the existence of this LD means that each selected site experiences an additional selective force caused by the other loci under selection. For site $i$, selection at site $j$ induces a change in allele frequency of $\delta q_{ij} = -sD_{ij}$ (Zhao and Charlesworth 2016). This corresponds to an additional selection coefficient at site $i$, which will affect the expected diversity at site, with $L-1$ contributions of this type from sites other than $i$. At first sight, this suggests that it is virtually impossible to obtain an expression for the expectation of $V_a$, since the $D_{ij}$ cannot easily be found analytically except for limiting cases such as complete neutrality. But the following argument shows that insights can be obtained without exact knowledge of the $D_{ij}$.

Simulations of blocks of nonrecombining, haploid loci show that HRI involves negative LD between deleterious mutations (e.g. McVean and Charlesworth 2000; Comeron and Kreitman 2002), implying that the effective selection coefficient against a deleterious mutation at a given site is reduced in magnitude compared with the selection coefficient in the absence of LD. This reduction provides an alternative perspective on HRI; the reduction in $N_e s$ caused by HRI is equivalent to this reduction in the effective selection coefficient (Zhao and Charlesworth 2016). However, inspection of Equation (2a) shows that the expected diversity at a site $i$ under selection ($\pi_{sel\ ij}$) is the product of $2u/s$ and a function of $N_e s$; only the former term need be considered further, since the effect of HRI on the latter is absorbed into $N_e s$.

A small alteration $\delta s$ in $s$ results in an approximate change of $(\delta s)\pi_{sel\ ij}/s$ in $\pi_{sel\ i}$, where $s$ is the selection coefficient in the absence of HRI effects. From the selection equation $\Delta q \approx -spq$, the change in $s$ at site $i$ due to selection at site $j$ is $\delta s_{sel\ ij} = -sD_{ij}/(p_i q_i)$, where $q_i$ is the frequency of the deleterious allele at site $i$ and $p_i = 1-q_i$ since the corresponding change in allele frequency at site $i$ is $-sD_{ij}$, as stated in the previous paragraph. The expected change in diversity at site $i$ due to selection at another site $j$ is thus:

$$E\{\delta\pi_{sel\ ij}\} \approx -\frac{2}{s}E\{p_i q_i\ \delta s_{sel\ ij}\} = -2E\{D_{ij}\} \tag{A1a}$$

The corresponding expected change in the genic variance at site $i$ is given by the product of $s^2$ and one-half of this quantity:

$$E\{\delta V_{gij}\} \approx -E\{D_{ij}\}s^2 \tag{A1b}$$

The net change in the genic variance at site $i$ is given by summing over all $j \neq i$.

There is an identical expected change to the genic variance at site $j$ due to selection at site $i$, $E\{\delta V_{gji}\}$. The corresponding contribution to the net expected additive genetic variance arising from LD between sites $i$ and $j$ is thus $2E\{D_{ij}\}s^2 = -E\{\delta V_{gij}\} - E\{\delta V_{gji}\}$. If second-order terms in $\delta s$ are neglected, the effect of LD associated with HRI on the sum of the genic variances across all sites approximately cancels the effect of the LD-induced covariance terms on the total additive genetic variance, given by the sum of $2\,D_{ij}s^2$ over all pairs of sites $i$ and $j$.

## 2. The neutral SFS

Let $f(i)$ be the probability that a derived neutral variant at a site is present in $i$ copies in a sample of size $k$. Knowledge of $f(i)$ provides an alternative method of determining $\Delta\theta_w$, without using any information about overall diversity statistics. Using the correction for bias in estimating pairwise diversity (Nei 1987, p. 178), the conditional pairwise diversity at segregating sites is given by:

$$\pi_c = \frac{2}{k(k-1)}\sum_{i=1}^{k-1} i(k-i)f(i) \tag{A2a}$$

If $P_s$ is the probability that a site is segregating, the theoretical values of $\theta_w$ and unconditional pairwise diversity are given by $\theta_w = P_s/a_k$ and $\pi = P_s\pi_c$. It follows that:

$$\Delta\theta_w = 1 - \pi_c a_k \tag{A2b}$$

If $k = 2$, only $i = 1$ contributes to Equation (A2a), so that $\pi_c = 1$; in this case, $a_k = 1$ and so $\Delta\theta_w = 0$. If the sample contains only singletons ($i = 1$), for $k > 2$ the maximum value of $\Delta\theta_w$ is given by:

$$\Delta\theta_{wm} = 1 - \frac{2a_k}{k} \tag{A2c}$$

The ratio $\Delta\theta_w/\Delta\theta_{wm}$ gives a standardized measure of the degree of distortion of the SFS toward rare variants, which we denote by $\Delta\theta'_w$.

*Editor: D. Roze*