## [Peer Review File · Genetics]

A model of Hill-Robertson interference caused by purifying selection in a non-recombining genome

Hannes Becher and Brian Charlesworth

NOTE: The reviews and decision letters are unedited and appear as submitted by the reviewers.

In extremely rare instances and as determined by a Senior Editor or the EIC, portions of a review may be redacted. If a review is signed, the reviewer has agreed to no longer remain anonymous.

The review history appears in chronological order.

Review Timeline:

Submission Date:	2024-11-04
Editorial Decision:	2024-12-13
Resubmission Received:	2025-02-04
Editorial Decision:	2025-03-06
Resubmission Received:	2025-03-11
Accepted:	2025-03-13

December 12, 2024

GENETICS-2024-307599

A model of Hill-Robertson interference caused by purifying selection in a non-recombining genome

Dear Dr. Becher:

Two experts in the field have reviewed your manuscript, and I have read it as well. While your manuscript is not currently acceptable for publication in GENETICS, we would welcome a substantially revised manuscript. You can read the reviewers' comments at the end of this email. In particular, reviewer 2 asks for a better justification of the biological significance of the model, and also has several criticisms concerning the presentation of the results, which could indeed be improved. This could perhaps be done by using the same values of gamma in table 2 and figure 4, and show the simulation results on the figure. Both reviewers also have several requests for clarification, and I have a few additional comments (at the end of this email).

We look forward to receiving your revised manuscript. Please let the editorial office know approximately how long you expect to need for revisions.

Upon resubmission, please include:

1. A clean version of your manuscript;
2. A marked version of your manuscript in which you highlight significant revisions carried out in response to the major points raised by the editor/reviewers (track changes is acceptable if preferred);
3. A detailed response to the editor's/reviewers' feedback and to the concerns listed above. Please reference line numbers in this response to aid the editor and reviewers.

Your paper will likely be sent back out for review.

Additionally, please ensure that your resubmission is formatted for GENETICS
<https://academic.oup.com/genetics/pages/general-instructions>

Follow this link to submit the revised manuscript: Link Not Available

Sincerely,

Denis Roze
Associate Editor
GENETICS

Approved by:
Nicholas Barton
Senior Editor
GENETICS

Reviewer #1 :

Comments by E. Santiago:

I reveal my identity in order to provide context for my comments on the theoretical aspects derived from Santiago and Caballero (2016). While I recognize the value of Becher and Charlesworth's contribution, it is refreshing to see my preferred work finally receiving some attention in the domains of background selection.

This work effectively accounts for the HRI effect on variation at neutral and selected sites in a model with reversible mutations without recombination, which evolves to a quasi-steady state distribution. The model is particularly useful for understanding the observed distribution of synonymous codon usage. I find the model useful, with accurate predictions of average frequency, B , B' , diversity, and site frequency spectrum (SFS). The work makes a valuable contribution to the understanding of HRI effects in non-recombining genomes.

While I was able to follow most of the manuscript, Appendix 1 and its related part in the main text were challenging. These are my comments:

L183 - The inclusion of B in "B gamma >> 1" seems unnecessary, since gamma is already proportional to Ne.

L204 vs. L217 -I think that pi_sel and pi_s are the same thing.

L214 - The statement about Va being equated to Vg, followed by the statement that LD increases Vg while decreasing Va by about the same amount, is confusing to the reader. Are Vg and Va the same or not?

Appendix 1 (following with the previous comment) - The Equation(2a) has two terms: $2u/s$, which is the expectation in infinitely large populations, and a finite size correction term. It sounds reasonable to include the HRI effect in both terms by reducing s in the first term and reducing Nes in the second. However, the argument for symmetric effects on Va and Vg is unclear. Does equation A1a suggest that an infinitesimal change in s leads to a small change in diversity equal to twice the total expected covariance? This seems implausible

L216 - The sentence "It follows..." seems to suggest that the underestimation of pi_s is a consequence of the claimed symmetry of changes in Vg and Va. If so, I disagree. In fact the sentence seems to contradict the development of the model in paragraph L312, which implicitly assumes that the difference between the "heterozygosity population size" and the Ne that determines the fixation probabilities (and hence the Va) is due to the increasing drift over the lifetime of the allele copy due to the cumulative effect of selection over generations at linked sites (Q).

L273- The symbol delta is introduced here and then abandoned three lines later. Does the delta mean that the value of ln(B) is infinitesimally small? Could delta be replaced by -ln(B) in Equation(8a)?

L355- I think that the year for the Polanski et al. paper is 2003.

L394 - The sentence about fixation "i.e., was present in 50% of the genomes" is confusing. The procedure for adapting the simulation programme to the haploid model is irrelevant here.

L729 - Equation (6) in Santiago and Caballero (2016) shows how to implement any distribution of mutation effects in the model (note: there is an obvious typo; it should include the f(s) in the integral). This equation was used to generate Figure S1 of our paper: the consequences of different distributions of effects on B. The conclusion was that the mean of the distribution captures most of the effects in sexual reproduction but not in asexuals, which is your case.

L731- The sentence "sites under strong selection affects sites under weak selection, but not vice-versa" does not accurately reflect reality. It is not all or nothing: weak sites do affect strong sites, but to a lesser extent. Because strong sites segregate for short periods of time, the cumulative effect of weak selection acting on linked sites is shorter and consequently the cumulative effect of selection is shorter. But strong deleterious sites also contribute little to the reduction of Ne for the same reason. The weak-strong reciprocal effects on Ne (and HRI) are accounted for in Equation(6) of Santiago and Caballero (2016).

L942 - Use lower case for the l.

Table 1- All the values should be presented with the same precision.

Table 2 - Address the following issues:

- Missing gamma value in the bottom row.
- Missing values of p in the second column.
- Failed numerical integration for the coalescent SFS prediction in the last row: a solution should be found using a different integration method.

Figure 3 - The distribution of neutral allele frequencies is actually symmetrical in frequency. If the x-axis represents a scale from 1 to 19 with the lowest frequency at 19 (i.e. the distribution appears to be folded), then the sample size must be ~40 alleles.

E. Santiago

Reviewer #2 :

Review of Becher and Charlesworth

In this manuscript the authors extend a theoretical framework to study the effects of Hill-Robertson Interference (HRI) due to deleterious alleles in the context of no recombination. The authors develop or extend a number of approximations for the scenario to derive quantities such as the equilibrium frequency of deleterious alleles, the effective population size relevant for fixation rate, diversity of neutral sites, and the site frequency spectrum.

Speaking in generalities, I have a number of issues with the current paper. These are both with respect to its purpose and its execution. For the former: why is a no recombination model of biallelic, recurrent mutation relevant? The authors seem to pitch this around optimal codon usage to some degree but as they themselves acknowledge, any real world non recombining chromosome will also be subject to large effect mutations (e.g. at non synonymous sites) that are not modeled here. That is true for Y chromosomes as well as neo-X chromosomes or mtDNA. On the execution side-the approximations derived aren't particularly accurate over the explored parameter space, nor are the results well presented (more on that later). Further the results are mostly confirmatory - there isn't a lot here that we didn't know before.

Some specific points:

1. It's not clear why one should care about a biallelic, recurrent model outside of say a codon usage framework. I'm unconvinced that this model is relevant to non-recombining portions of real genomes, particularly in light of the fact that we know that gene conversion still remains an important force in those portions of the genome (e.g. the 4th chromosome of *Drosophila*).
2. The presentation of the results in this paper is a mess. Table 2, which contains the bulk of the important comparisons between the approximations and the simulations is close to unreadable. For instance CIs are given for some quantities but not others. CIs are broken across lines. Comparisons between simulation and approximation are not provided in different columns but across rows in the same cell. The value of Gamma is wholly missing from the last row. Etc. I found myself having to go back and forth from the poorly formatted table legend the table itself just to make sense of what is happening here. These results would be much better presented in simple figures. The Figures as well are difficult to follow. For instance in Figure 2 no legend is given with respect to the markers. Figure 4 is likewise not well presented-in this figure the authors are showing results from 5 different quantities with no proper legend and a bizarrely chosen x-axis scale.
3. The supplemental materials in Supps 2 and 3 are not helpful. How is one supposed to read SM2? What are the tables given in SM3 and how are they meant to provide useful information for the reader? Generally this was quite unpolished.
4. The section of the manuscript that deals with corrections for LD (lines 459-468) was very unclear. All of this should be laid out clearly and not with verbal descriptions such as "the product of $-4/L(L-1)$ and the sum of the Ds".
5. While the authors have done a good job generally with the scholarship throughout the manuscript, there is a glaring omission here. A now 6 month old preprint from Strutt et al. (<https://doi.org/10.1101/2024.06.11.598434>) develops approximations for a very similar scenario. These should be compared to the authors' results. I was also surprised that the authors didn't compare their SFS results to those of Cvijovic et al (2018) who deal with a strong selection regime.

Some minor points:

1. Line 61 - fixations -> fixation
2. L68 - referencing Maynard Smith and Haigh here is a strange choice. There is not a model of HRI which specifically deals with the interference interactions among selected loci.
3. L 217 - type. Should be π_{sel} I believe
4. L528-531. I don't understand the relevance of the comparison here. Are we supposed to believe there is no HRI on complete chromosome arms? Or that the model would be an adequate description of this biological entity? Further in L532 the authors talk about B declining to 0.1 - there is nearly no portion of the genome that shows B at that level from previous estimates
5. L611-621 - given the level of precision of the approximations this section of the discussion is not convincing
6. A small note that in the simulation code, comments like "// absence of an m2 mutation is 'a'; 'aa' is neutral" make it sound like the model is diploid when in fact it is haploid. I'll also note that in running the simulations, mean population fitness was still decreasing at the end of the simulation-while it was close to achieving stable state it was not there yet.

Associate Editor Comments:

The result that the fixation and coalescence effective size (is the former equivalent to the variance effective size?) is of interest and could perhaps be developed further. In particular, I wondered if this difference is due to the fact that background selection changes the topology of the coalescence tree.

I.42-46: the assumption of reversibility could rather be justified by the fact that you are considering weakly selected deleterious alleles, that may fix with non-negligible probabilities. Previous works have shown that neglecting back mutation does not affect much the results when $N_e s \gg 1$, so that deleterious alleles stay close to deterministic mutation-selection balance.

I.130 and 131-132 seem contradictory

I.223: remove among or across

I.363-379: more explanations about what Delta theta w is supposed to measure would be helpful

Table 1: eq.6 does not seem to yield 0.741 for those parameter values?

Responses to the Reviews

Reviewer #1 :

Comments by E. Santiago:

I reveal my identity in order to provide context for my comments on the theoretical aspects derived from Santiago and Caballero (2016). While I recognize the value of Becher and Charlesworth's contribution, it is refreshing to see my preferred work finally receiving some attention in the domains of background selection.

This work effectively accounts for the HRI effect on variation at neutral and selected sites in a model with reversible mutations without recombination, which evolves to a quasi-steady state distribution. The model is particularly useful for understanding the observed distribution of synonymous codon usage. I find the model useful, with accurate predictions of average frequency, B , B' , diversity, and site frequency spectrum (SFS). The work makes a valuable contribution to the understanding of HRI effects in non-recombining genomes.

We thank Prof. Santiago for his kind remarks. We have tried to address his detailed comments, as explained below.

While I was able to follow most of the manuscript, Appendix 1 and its related part in the main text were challenging. These are my comments:

L183 - The inclusion of B in " $B \gamma \gg 1$ " seems unnecessary, since γ is already proportional to N_e .

This has been corrected to $B\gamma_0$.

L204 vs. L217 - I think that π_{sel} and π_s are the same thing.

This has been corrected.

L214 - The statement about V_a being equated to V_g , followed by the statement that LD increases V_g while decreasing V_a by about the same amount, is confusing to the reader. Are V_g and V_a the same or not?

V_g is the sum of the variances at each site; V_a is the net variance, which includes a covariance term arising from LD (Crow and Kimura and/or Bulmer). We have now added an explicit equation for V_a that includes the LD component (1.223-225 of the revision).

Appendix 1 (following with the previous comment) - The Equation(2a) has two terms: $2u/s$, which is the expectation in infinitely large populations, and a finite size correction term. It sounds reasonable to include the HRI effect in both terms by reducing s in the first term and reducing $N_e s$ in the second. However, the argument for symmetric effects on V_a and V_g is unclear. Does equation A1a suggest that an infinitesimal change in s leads to a small change in diversity equal to twice the total expected covariance? This seems implausible.

The reviewer has overlooked the fact that summing $n(n-1)/2$ small terms each involving a very small pairwise D can give you a substantial net term (e.g. Bulmer 1980, p.158).

L216 - The sentence "It follows..." seems to suggest that the underestimation of π_s is a consequence of the claimed symmetry of changes in V_g and V_a . If so, I disagree. In fact the sentence seems to contradict the development of the model in paragraph L312,

which implicitly assumes that the difference between the "heterozygosity population size" and the N_e that determines the fixation probabilities (and hence the V_a) is due to the increasing drift over the lifetime of the allele copy due to the cumulative effect of selection over generations at linked sites (Q).

The reviewer appears to equate diversity at selected sites with neutral diversity; our point is that the effect of selection at linked sites on the former is to reduce s in the denominator of Eqn. 2a, causing an increase in π_{sel} , as well as reducing γ , so that ignoring the former effect leads to an underestimation of π_{sel} . This is not related to the effect of selection on neutral diversity, which we analyse using the coalescent process (1.310-358). We have inserted a clarification of the relation between the two different measures of N_e on 1.337-341.

L273- The symbol delta is introduced here and then abandoned three lines later. Does the delta mean that the value of $\ln(B)$ is infinitesimally small? Could delta be replaced by $-\ln(B)$ in Equation(8a)?

We have clarified the fact that Eqns. (8) are valid only when δ is sufficiently small that second-order terms in it can be neglected (1.287-288), and now mention them on 1.294.

L355- I think that the year for the Polanski et al. paper is 2003.

This error has been corrected.

L394 - The sentence about fixation "i.e., was present in 50% of the genomes" is confusing. The procedure for adapting the simulation programme to the haploid model is irrelevant here.

We have removed this statement.

L729 - Equation (6) in Santiago and Caballero (2016) shows how to implement any distribution of mutation effects in the model (note: there is an obvious typo; it should include the $f(s)$ in the integral). This equation was used to generate Figure S1 of our paper: the consequences of different distributions of effects on B . The conclusion was that the mean of the distribution captures most of the effects in sexual reproduction but not in asexuals, which is your case.

We are somewhat sceptical about the accuracy of the Santiago and Caballero Equation (6). The integration involves the variances for each selective class; it is not obvious that the same N_e will apply to all such classes, as is tacitly assumed there. We feel that more validation by simulation is needed before we can have confidence in this expression, but this is beyond the scope of the present paper.

L731- The sentence "sites under strong selection affects sites under weak selection, but not vice-versa" does not accurately reflect reality. It is not all or nothing: weak sites do affect strong sites, but to a lesser extent. Because strong sites segregate for short periods of time, the cumulative effect of weak selection acting on linked sites is shorter and consequently the cumulative effect of selection is shorter. But strong deleterious sites also contribute little to the reduction of N_e for the same reason. The weak-strong reciprocal effects on N_e (and HRI) are accounted for in Equation(6) of Santiago and Caballero (2016).

We have modified our statement to take this interesting remark into account (1.804-808); however, it should be noted that there are more nonsynonymous than synonymous sites in coding sequences which partially counteracts the effects of shorter sojourn times referred to

here. The previous work on the ratchet that we cite showed that there can be a strong asymmetry when there are large differences in selection coefficients.

L942 - Use lower case for the I.

Corrected.

Table 1- All the values should be presented with the same precision.

Done

Table 2 - Address the following issues:

As noted in the response to reviewer 2, we have now split these results between a new Figure 3 and a smaller Table 2, which we believe makes for greater clarity.

- Missing gamma value in the bottom row.

Corrected.

- Missing values of p in the second column.

Corrected.

- Failed numerical integration for the coalescent SFS prediction in the last row: a solution should be found using a different integration method.

A new value has been determined using the same method.

Figure 3 - The distribution of neutral allele frequencies is actually symmetrical in frequency. If the x-axis represents a scale from 1 to 19 with the lowest frequency at 19 (i.e. the distribution appears to be folded), then the sample size must be ~ 40 alleles.

The figure shows the unfolded site frequency spectrum, i.e, the distribution of the number of derived variants. This is not symmetrical; the expectation under strict neutrality with constant population size is that i derived variants should be present in a sample of size n with a probability proportional to $1/i$, where i runs from 1 to $n - 1$ (see texts such as Ewens 2004 or Wakeley 2008).

Reviewer #2 :

Review of Becher and Charlesworth

In this manuscript the authors extend a theoretical framework to study the effects of Hill-Robertson Interference (HRI) due to deleterious alleles in the context of no recombination. The authors develop or extend a number of approximations for the scenario to derive quantities such as the equilibrium frequency of deleterious alleles, the effective population size relevant for fixation rate, diversity of neutral sites, and the site frequency spectrum.

Speaking in generalities, I have a number of issues with the current paper. These are both with respect to its purpose and its execution. For the former: why is a no

recombination model of biallelic, recurrent mutation relevant? The authors seem to pitch this around optimal codon usage to some degree but as they themselves acknowledge, any real world non-recombining chromosome will also be subject to large effect mutations (e.g. at non-synonymous sites) that are not modeled here. That is true for Y chromosomes as well as neo-X chromosomes or mtDNA. On the execution side-the approximations derived aren't particularly accurate over the explored parameter space, nor are the results well presented (more on that later). Further the results are mostly confirmatory - there isn't a lot here that we didn't know before.

We feel that the reviewer is not entirely fair to our work. We cite some recent *Genetics* papers concerned with exactly this topic – the effects of selection on the properties of linked sites in a non-recombining genome. Most of these also simplify the problem by assuming a fixed selection coefficient across sites and use much more complex modelling approaches. Does the reviewer believe that these earlier papers should not have been published?

We certainly agree with the reviewer that considering the effects of sites with different selection coefficient is important, but an exact treatment is extremely hard to achieve (see discussion on 1.797-810). It is surely of value to show that even a quite modest number of quite weakly selected sites can interfere with each other and reduce the level of adaptation and diversity in a low recombination genome, as we do here.

In addition, we show how to correct for the effects of LD on diversity at selected sites, a problem that was overlooked in previous work employing the Santiago & Caballero approach. We also provide a new and quite accurate approach to calculating the neutral site frequency spectrum for our model, which we believe is a substantial advance on previous work. We have a final section that discusses the various limitations of the work, which has now been slightly expanded (1.800-812).

We also note that reviewer 1 had a different view of the merit of our work.

Some specific points:

1. It's not clear why one should care about a biallelic, recurrent model outside of say a codon usage framework. I'm unconvinced that this model is relevant to non-recombining portions of real genomes, particularly in light of the fact that we know that gene conversion still remains an important force in those portions of the genome (e.g. the 4th chromosome of *Drosophila*).

The importance of considering reversible mutation is that large numbers of selected sites in a non-recombining genome can have a major effect in reducing the mean frequency of the favoured allele at even quite strongly selected sites to intermediate values (see Table 1, for example). Thus, the results apply far beyond the codon usage framework. We have added brief statements about this to 1.76-79 and 1.142-150.

We would argue that, in fact, ignoring reversible mutation is inappropriate when considering strong HRI effects except for types of mutations that are essentially irreversible (e.g. those considered by Kaiser & Charlesworth 2010). Does anybody really believe that mean fitness of non-recombining populations declines indefinitely over time, as shown in papers like Strütt et al. (2024, Fig.1)? See Charlesworth et al. (2010) for a discussion of this issue.

The reviewer's remark about gene conversion on the 4th chromosome is correct (see 1.813-815); however, one application of our model is to Y or W chromosomes in diploids. In

species with no recombinational exchange in the heterogametic sex, such as *Drosophila* or *Lepidoptera*, there is a total absence of homologous recombination, so this criticism does not apply to such systems. In addition, by continuity a low level of exchange in a diploid should not greatly affect the overall results – Kaiser and Charlesworth (2009, Fig 2) showed that a realistic rate of gene conversion had little effect on their simulation results.

2. The presentation of the results in this paper is a mess. Table 2, which contains the bulk of the important comparisons between the approximations and the simulations is close to unreadable. For instance CIs are given for some quantities but not others. In simulation and approximation are not provided in different columns but across rows in the same cell. The value of Gamma is wholly missing from the last row. Etc. I found myself having to go back and forth from the poorly formatted table legend the table itself just to make sense of what is happening here. These results would be much better presented in simple figures.

We agree with this criticism; Table 2 was botched. We have taken the results on \bar{p} , B and B' out of Table 2 and made a new Fig. 3 for these, which we believe makes them easy to understand. The new version of Table 2 has been corrected for the errors noted by the reviewer.

The Figures as well are difficult to follow.
For instance in Figure 2 no legend is given with respect to the markers.

Figs 1 and 2 have been redone.

Figure 4 is likewise not well presented-in this figure the authors are showing results from 5 different quantities with no proper legend and a bizarrely chosen x-axis scale.

Fig 4 has been redone (now Fig 5).

3. The supplemental materials in Supps 2 and 3 are not helpful. How is one supposed to read SM2? What are the tables given in SM3 and how are they meant to provide useful information for the reader? Generally this was quite unpolished.

We have removed the offending material in Supps S2 and S4-S6, which were intended as data-dumps, and placed it on Zenodo along with the computer code. We now have Supplementary Files S2 and S4, which contain the numbers used to generate Figs. 1, 2 and 5. S3 has been revised to make it more user-friendly.

4. The section of the manuscript that deals with corrections for LD (lines 459-468) was very unclear. All of this should be laid out clearly and not with verbal descriptions such as "the product of $-4/L(L-1)$ and the sum of the D_s ".

An algebraic treatment now replaces the verbal account – see 1.482-493.

5. While the authors have done a good job generally with the scholarship throughout the manuscript, there is a glaring omission here. A now 6 month old preprint from Strutt et al. (<https://doi.org/10.1101/2024.06.11.598434>) develops approximations for a very similar scenario. These should be compared to the authors' results.

We were aware of this paper. It was not a peer-reviewed publication, so we had reservations about citing it, but it is now Advance Online in *Genetics*. We now include brief citations

(1.74, 370); it's not strictly comparable to ours as it doesn't include reversible mutations, nor do they explicitly present SFS results. Like ours, they assume a single selection coefficient.

I was also surprised that the authors didn't compare their SFS results to those of Cvijovic et al (2018) who deal with a strong selection regime.

We unfortunately forgot to cite this paper. Our paper also embraces a strong selection (more correctly, a large U/s regime), and we also detect distortions of the SFS, even in relatively small samples. Our simulations confirm the existence of a small uptick in the probability of high frequency derived variants which they predict, even for small U/s (see 1.780-792). We believe that our approach is more realistic, since we avoid the assumption of one-way mutations. The main drawback with our approach is that the SFS for large samples becomes hard to compute with the Polanski-Kimmel approach and does not predict the uptick for the high frequency classes.

We now cite their work on 1.73, 139, 316, 610, 785.

Some minor points:

1. Line 61 - fixations -> fixation

Corrected.

2. L68 - referencing Maynard Smith and Haigh here is a strange choice. Theirs is not a model of HRI which specifically deals with the interference interactions among selected loci.

Replaced with Kim (2004) and Weissman & Barton (2012) – 1.57.

3. L 217 - type. Should be π_{sel} I believe

Corrected.

4. L528-531. I don't understand the relevance of the comparison here. Are we supposed to believe there is no HRI on complete chromosome arms? Or that the model would be an adequate description of this biological entity?

We should have made it clearer that we had in mind a neo-Y chromosome like that of *D. miranda*, where a whole chromosome arm that was previously recombining at a high rate becomes non-recombining. We have rewritten this entire paragraph to clarify this (1.550-562).

Further in L532 the authors talk about B declining to 0.1 - there is nearly no portion of the genome that shows B at that level from previous estimates

This is incorrect; we give examples from the *Drosophila* literature that show that B' can be as low as 1% (1.556-559).

5. L611-621 - given the level of precision of the approximations this section of the discussion is not convincing

We did put in a caution about this (1.643-646). We now add that the analytical results provided a conservative estimate of the magnitude of this measure of the reduced efficacy of selection (1.656-660), and hence are quite useful.

6. A small note that in the simulation code, comments like "// absence of an m2 mutation is 'a'; 'aa' is neutral" make it sound like the model is diploid when in fact it is haploid. I'll also note that in running the simulations, mean population fitness was still decreasing at the end of the simulation-while it was close to achieving stable state it was not there yet.

Regarding the simulation code comments - diploidy is hard-coded into SLiM, which is why our comments refer to SLiM's diploid computational model. However, we have set up our simulations in such a way that SLiM behaves as if individuals were haploid.

Thank you for making the point about the equilibrium state. We have now fitted asymptotic curves to the data from our log files and it seems that indeed, the average allele frequency values had not quite reached the asymptotic value for some parameter sets. We also note that the maximum deviation was a deficit 1.7% compared to the asymptotic value. This was for the parameter set with $L = 40,000$ and $s = 1/1000$. We do not think that this significantly affects our conclusions, since the deviations are so small.

Associate Editor Comments:

The result that the fixation and coalescence effective size (is the former equivalent to the variance effective size?) is of interest and could perhaps be developed further. In particular, I wondered if this difference is due to the fact that background selection changes the topology of the coalescence tree.

This was discussed by Santiago & Caballero (2016). It comes about because the fixation N_e reflects the cumulative effects of the fitness variance on the rate of drift over a long period of time, whereas the effects on diversity are on a shorter time scale. Equation (4) for the fixation N_e is the asymptotic value of $N_e(\tau)$ in Equations (10). We have inserted a comment to this effect on 1.337-341.

1.42-46: the assumption of reversibility could rather be justified by the fact that you are considering weakly selected deleterious alleles, that may fix with non-negligible probabilities. Previous works have shown that neglecting back mutation does not affect much the results when $Ne s \gg 1$, so that deleterious alleles stay close to deterministic mutation-selection balance.

We have discussed this issue in the reply to specific comment 1 of reviewer 2 (see above). The comment overlooks the fact that HRI itself can cause strongly selected sites to become nearly neutral.

1.130 and 131-132 seem contradictory

There was a typo that we have now corrected.

1.223: remove among or across

Corrected.

1.363-379: more explanations about what Delta theta w is supposed to measure would be helpful

1st Revision - Authors' Response to Reviewers: February 4, 2025

An explanation is inserted on l.391-395, citing Tajima (1989) who first pointed this out.

Table 1: eq.6 does not seem to yield 0.741 for those parameter values?

We have checked this and get the same result.

March 6, 2025

GENETICS-2025-307844

A model of Hill-Robertson interference caused by purifying selection in a non-recombining genome

Dear Dr. Becher:

Two experts in the field have reviewed your manuscript, and I have read it as well. I am pleased to inform you that, with minor revisions, it is potentially suitable for publication in GENETICS. Indeed, reviewer 1 still has a number of (relatively minor) comments that should be addressed.

We look forward to receiving your revised manuscript. Please let the editorial office know approximately how long you expect to need for revisions.

Upon resubmission, please include:

1. A clean version of your manuscript;
2. A marked version of your manuscript in which you highlight significant revisions carried out in response to the major points raised by the editor/reviewers (track changes is acceptable if preferred);
3. A detailed response to the editor's/reviewers' comments and to the concerns listed above. Please reference line numbers in this response to aid the editors.

Additionally, please ensure that your resubmission is formatted for GENETICS.

<https://academic.oup.com/genetics/pages/general-instructions>

Follow this link to submit the revised manuscript: Link Not Available

Sincerely,

Denis Roze
Associate Editor
GENETICS

Approved by:
Nik Barton
Senior Editor
GENETICS

Reviewer #1 :

Comments on the manuscript by Becher and Charlesworth:

I have only four minor comments regarding some inaccuracies that should be addressed:

Line 197: The statement that the approach to modeling variance at a single selected locus "differs from the approach of Santiago & Caballero 2016" seems inaccurate. Both the current manuscript and S&C use equations based on Kimura's probability of fixation in a single-locus model. The novelty of this manuscript is the consideration of reversible mutations.

Line 272: The first sentence of this paragraph incorrectly implies that the S&C 2016 model is equivalent to the classical BGS model. In fact, the S&C model accounts for departures from mutation-selection equilibrium. Please move the S&C 2016 citation to the final sentence of the paragraph for clarity.

Line 805: The skepticism expressed about the S&C model's ability to handle continuous selective effects seems to be based on studies of the classical BGS model, which is indeed limited in this regard. This argument is not robust enough. If you believe that further simulations are needed to validate the capabilities of the S&C model, it would be advisable to present this as an open question for future research.

Line 1048: The statement "HRI induces negative LD between deleterious mutations" misrepresents the causal relationship.

More precisely, negative LD induces HRI. If you manipulate genotypes to produce negative LD, you will have HRI when you select. The correct sequence is: selection acting on individuals -> negative LD between gene effects in the selected group -> HRI in future gene frequency changes.

E. Santiago

Reviewer #2 :

this version of the paper is in better shape than the original and i'm pleased with the responses the authors have provided

Associate Editor Comments:

Reviewer #1 :

Comments on the manuscript by Becher and Charlesworth:

I have only four minor comments regarding some inaccuracies that should be addressed:

Thank you for your continued constructive criticism. We have acted on your suggestions.

Line 197: The statement that the approach to modeling variance at a single selected locus "differs from the approach of Santiago & Caballero 2016" seems inaccurate. Both the current manuscript and S&C use equations based on Kimura's probability of fixation in a single-locus model. The novelty of this manuscript is the consideration of reversible mutations.

We agree with you that one novel aspect of this manuscript is that we consider reversible mutations. The paragraph which you are referring to does state that we use "a value of N_e reflecting the effects of HRI". This is a critical difference from Santiago & Caballero 2016 and the other works that we have referenced. We have now added the words 'subject to reversible mutation and genetic drift' (l.206-7) and 'importantly' (l.208) to the subclause about N_e .

Line 272: The first sentence of this paragraph incorrectly implies that the S&C 2016 model is equivalent to the classical BGS model. In fact, the S&C model accounts for departures from mutation-selection equilibrium. Please move the S&C 2016 citation to the final sentence of the paragraph for clarity.

We have moved the reference as requested (l.289-290).

Line 805: The skepticism expressed about the S&C model's ability to handle continuous selective effects seems to be based on studies of the classical BGS model, which is indeed limited in this regard. This argument is not robust enough. If you believe that further simulations are needed to validate the capabilities of the S&C model, it would be advisable to present this as an open question for future research.

Thank you for pointing this out, we have now changed the wording to (additions in italics):

"integrating Equations (10) over a distribution of selection coefficients (Santiago and Caballero 2016; Buffalo and Kern 2024) may not be adequate in low recombination regimes, *a topic that should be explored in future simulation studies*. Our concern is that sites under strong selection are likely to have larger effects on weakly selected sites than vice-versa" (l.824)

Line 1048: The statement "HRI induces negative LD between deleterious mutations" misrepresents the causal relationship. More precisely, negative LD induces HRI. If you manipulate genotypes to produce negative LD, you will have HRI when you select. The correct sequence is: selection acting on individuals -> negative LD

between gene effects in the selected group -> HRI in future gene frequency changes.

We certainly agree with your reasoning that selection acting on individuals can induce negative LD between and that HRI is then observed. We have changed 'induces' to 'involves' (l.1070)

E. Santiago

Reviewer #2 :

this version of the paper is in better shape than the original and i'm pleased with the responses the authors have provided

Thank you.

March 13, 2025

RE: GENETICS-2025-307972

Dr. Hannes Becher
The University of Edinburgh
The Roslin Institute, Royal (Dick) School of Veterinary Studies
Midlothian
Edinburgh, N/A EH25 9RG
United Kingdom

Dear Dr. Becher:

Congratulations! We are delighted to inform you that your manuscript titled "A model of Hill-Robertson interference caused by purifying selection in a non-recombining genome" is acceptable for publication in GENETICS. Many thanks for submitting your research to the journal.

To Proceed to Production:

1. Format your article according to GENETICS style, as discussed at <https://academic.oup.com/genetics/pages/general-instructions>, and upload your final files at <https://genetics.msubmit.net>.
2. Your manuscript will be published as-is (unedited-as submitted, reviewed, and accepted) at the GENETICS website as an Advanced Access article and deposited into PubMed shortly after receipt of source files and the completed license to publish. Please notify sourcefiles@thegsajournals.org if you do not wish to publish your article via Advanced Access.
3. We invite you to submit an original color figure related to your paper for consideration as cover art. Please email your submission to the editorial office or upload it with your final files. You can submit a small-sized image for evaluation, and if selected, the final image must be a TIFF file 2513px wide by 3263px high (8.375 by 10.875 inches; resolution of 600ppi). Please avoid graphs and small type.

If you have any questions or encounter any problems while uploading your accepted manuscript files, please email the editorial office at sourcefiles@thegsajournals.org.

Sincerely,

Denis Roze
Associate Editor
GENETICS

Approved by:
Nicholas Barton
Senior Editor
GENETICS

note: Please add jnls.author.support@oup.com and genetics.oup@kwglobal.com (or the domains @oup.com and @kwglobal.com) to your email program's "safe senders" list. You will be contacted by both at various points during the production process.